# Compositional Generalization through Gradient Search in Nonparametric Latent Space

**Haruki Shirakami**
EPFL, Idiap Research Institute
haruki.shirakami@epfl.ch

**James Henderson**
Idiap Research Institute
james.henderson@idiap.ch

## Abstract

Many state-of-the-art methods in deep learning fail at systematic reasoning in settings which require compositional generalization. To address this, we propose a novel architecture which uses a nonparametric latent space, information-theoretic regularization of this space, and test-time gradient-based search to achieve strong performance on compositional meta-learning tasks such as program induction, Raven's progressive matrices, and linguistic systematicity tasks. Our proposed architecture, Abduction Transformer, uses nonparametric mixture distributions to represent inferred hidden causes of few-shot meta-learning instances. These representations are refined at test-time via gradient descent to better account for the observed few-shot examples, a form of variational posterior inference which allows Abduction Transformer to solve meta-learning tasks that require novel recombinations of knowledge acquired during training. Our method outperforms standard transformer architectures and a contemporary test-time adaptive variational approach, indicating a promising new direction for neural networks capable of systematic generalization.[1]

## 1 Introduction

The ability to solve novel tasks by recombining known primitives, often referred to as *compositional generalization*, is commonly understood as a prerequisite for general intelligence (Fodor, 1975; Cattell, 1963; Chomsky, 1965; Gick & Holyoak, 1980). In settings where exhaustive memorization of the solutions to problem instances is impossible, the ability to learn and compose reusable concepts is necessary for solving problems in general; any system which fails to do so will exhibit catastrophic failure on problems which are out of distribution (Chollet, 2019). Historically, it has been debated whether or not connectionist architectures such as neural networks possess the capacity to learn representations and circuits which allow for this form of systematic recombination of knowledge to infer new concepts (Rumelhart & McClelland, 1986; Smolensky, 1990; Fodor & Pylyshyn, 1988). While some progress has been made to refute this claim (Lake & Baroni, 2023), and despite the rise of LLMs which seemingly display some characteristics of general intelligence and compositional reasoning (Bubeck et al., 2023; An et al., 2023; Hosseini et al., 2022), a plethora of negative results suggest that current architectural paradigms are inadequate and so further investigation is warranted (Dziri et al., 2023; Mirzadeh et al., 2025; Opedal et al., 2025; Shojaee et al., 2025).

In this work, we propose a novel architecture with strong compositional generalization ability. We demonstrate the presence of these abilities in our proposed methods by considering meta-learning tasks which necessitate out-of-distribution (OOD) adaptation to novel test instances, requiring some form of knowledge recombination. In particular, we study program induction (Summers, 1977; Biermann, 1978) and grammar induction (Lake & Baroni, 2023), abstract meta-learning tasks which standard transformer architectures perform poorly on due to their compositional nature. While some combinations of compositional generalization (Schug et al., 2025; Chen et al., 2020), test-time adaptation (Hübotter et al., 2025; Dong et al., 2025; Gladstone et al., 2025; Mathur et al., 2025), meta-learning (Vettoruzzo et al., 2025; Yao et al., 2022), and abstract reasoning (Wang et al., 2025; Li et al., 2024) have each been studied in isolation, we are, to our knowledge, the first to propose and evaluate a method covering all of these aspects.

---

[1]Code will be published at  https://github.com/idiap/AbductionTransformer.

Problems of this type require some form of search over hypotheses inferred from observations and prior knowledge (Macfarlane & Bonnet, 2025; Chollet et al., 2025). In our case, these hypotheses include possible combinations of concepts and abstractions learned during training. By representing these hypotheses in some latent space, the problem of knowledge recombination (and by extension *compositional reasoning*) becomes a matter of discovering some latent representation of the novel input which allows it to be explained using the generative model acquired during training, giving us a form of hypothesis testing.

Inferring such a latent cause of the observed input is abductive inference, so we call our proposed architecture **Abduction Transformer**. Abduction Transformer is a deep variational Bayesian model, like Variational Autoencoders (Kingma et al., 2013) and Latent Program Networks (Macfarlane & Bonnet, 2025), but unlike these previous models it takes full advantage of the power of the transformer architecture's set-of-vector embeddings by encoding the hidden causes as *nonparametric* latent representations (Henderson & Fehr, 2023). Nonparametric representations have the advantage that they generalize well across situations of varying complexity, such as generalizing from simple concepts learned during training to their more complex compositions. Crucially, our method includes not only amortized inference but also test-time gradient-based search over its latent space to find the most plausible hypotheses which account for problem instances. We show that this test-time search procedure, combined with our information-theoretic regularization over our latent space, enables discovery of minimal latent representations of inputs and consequently allows our model to perform well on novel tasks.

Our main contributions are summarized below:

i) **We find that set-of-vector representations with variable cardinality lead to better compositional generalization.** We represent inputs as nonparametric (variable-sized) discrete mixture distributions, as opposed to parametric (fixed-sized) vectors commonly used in previous Bayesian methods.

ii) **We show that test-time gradient search over latent representations leads to improved generalization in extreme OOD regimes.** Test-time search enables our models to solve problems containing a vast number of unseen concept combinations, which standard transformers with the same training data are unable to solve.

iii) **We demonstrate that stochastic sampling of representations at training-time leads to a searchable latent space.** By encoding inputs into parameters of Dirichlet processes from which we sample discrete mixtures, our latent space benefits from information-theoretic regularization, resulting in a smooth and *searchable* space.

iv) **Our models significantly outperform standard transformer architectures and previous test-time adaptive methods on OOD abstract reasoning tasks.** In addition, we nominally outperform GPT-5 Thinking (OpenAI, 2025) (w.o. fine-tuning) on Raven's progressive matrices (Raven, 1962) and perform comparatively on 1-D ARC problems (both in OOD settings), while using only ~1.2M parameters.

Overall, our contributions make significant progress towards developing test-time adaptive neural networks which are capable of knowledge recombination in novel situations.

## 2 META-LEARNING AS INFERRING HIDDEN MAPPINGS

**Problem definition.** We consider few-shot meta-learning problems where the objective is to infer some hidden mapping which explains the few-shot examples. The few-shot examples are defined as a set of input/output pairs, which form a problem specification:

$$X = \{(x_1, y_1), (x_2, y_2), ..., (x_n, y_n)\}. \tag{1}$$

A mapping $H^*$ is said to solve the problem if $H^*(x_i) = y_i, \forall i \in [1, n]$. In practice, we test whether the correct prediction of the ground truth test output $y^*$ on some test query $x_{\text{query}}$ is made, namely that $y^* = H^*(x_{\text{query}})$.

**Hypotheses and compositions.** We refer to a candidate mapping $H$ as a **hypothesis** coming from some hypothesis space $\mathcal{H}$. A problem specification $X$ is considered **function compositional** if its solution $H^*$ can be expressed as a composition of two mappings $H_1, H_2 \in \mathcal{H}$ such that $H^* = H_1 \circ H_2$. In addition, a problem specification with test query $x_{\text{query}}$ and ground truth test output

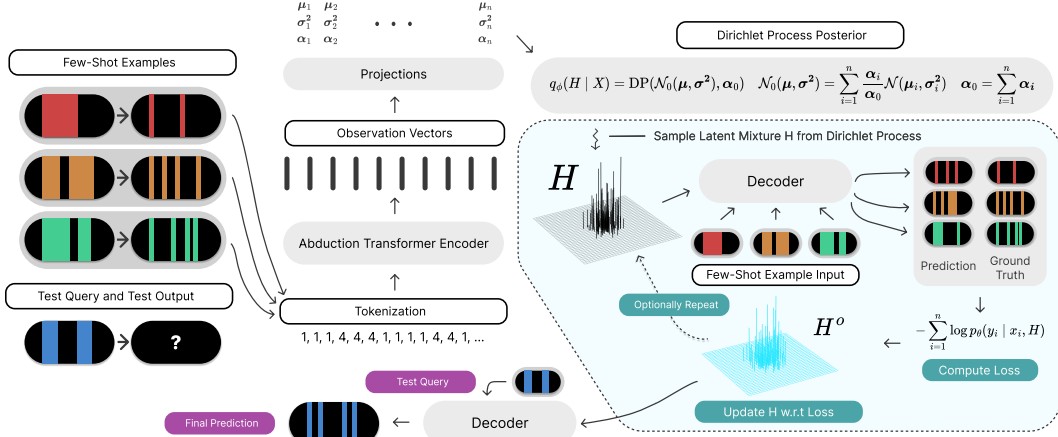

Figure 1: **Overview of Abduction Transformer. Left:** Few-shot example pairs are tokenized and encoded into a set of vectors from which the parameters of a posterior Dirichlet process are inferred. **Right:** A latent mixture distribution $H$ is sampled from the DP and used to decode the inputs of few-shot examples into predicted outputs, from which cross-entropy loss is computed against the true outputs. We update our latent representation $H$ w.r.t. this loss to refine our hypothesis to better account for the few-shot examples. After repeating this process a fixed number of times, we use our refined latent representation to decode the test query into the final prediction.

$y^*$ of the same form as Eq. 1 is considered **production compositional** if there exist mappings $H_1, H_2 \in \mathcal{H}$ such that:

$$y_i = H_1(x_i) \quad y_j = H_2(x_j) \quad y^* = H_1(H_2(x_{\text{query}})) \quad i, j \in [1, n] \tag{2}$$

We refer to the individual mappings constituting the compositions as **constituent hypotheses**. A tuple $(X, x_{\text{query}}, y^*)$ is said to be function compositional *relative to* a set of mappings $\mathcal{T}$ if $\mathcal{T}$ contains its constituent hypotheses, and the same can be said for production compositional problem instances. If a system $\mathcal{A}$ which has been trained to recognize a set of hypotheses $\mathcal{T} \subseteq \mathcal{H} \setminus \{H^*\}$ is able to solve a problem $X$ with solution $H^*$ which is function compositional or production compositional relative to $\mathcal{T}$, we say that system $\mathcal{A}$ exhibits **compositional generalization**.

## 3 ABDUCTION TRANSFORMER

**Few-shot learning as variational inference.** The objective of the few-shot meta-learning task presented in the previous section can be viewed as posterior inference of the hypothesis $H$ which best accounts for the observations specified by $X$. In other words, the task amounts to inferring the posterior distribution $p(H \mid X)$.

Fig. 1 gives an overview of the Abduction Transformer architecture. Similarly to VAEs (Kingma et al., 2013), we define parameterized neural networks which approximate the distributions $q_\phi(H \mid X)$ (encoder) and $p_\theta(X \mid H)$ (decoder), and train these networks to minimize variational free energy, an upper bound on the KL-divergence between the true posterior $p(H \mid X)$ and $q_\phi(H \mid X)$[2]:

$$\mathcal{F}(\phi, \theta) = \text{KL}(q_\phi(H \mid X) \,\|\, p(H)) - \mathbb{E}_{q_\phi(H|X)}[\log p_\theta(X \mid H)] \tag{3}$$

**A nonparametric latent space.** Our architecture uses a transformer encoder to infer $q_\phi(H \mid X)$, which has the crucial property that the number of vectors in the transformer's output grows proportionally to the number of tokens in its input. This ability to model situations of variable complexity is analogous to the way Bayesian nonparametrics is able to model mixture distributions of variable complexity. We take advantage of this ability to model mappings $H$ with variable complexity, so that models trained on simple mappings generalize naturally to their more-complex compositions.

To do so, we use an approach proposed by Henderson & Fehr (2023) for general purpose transformer VAEs. We project the transformer output into the parameters of a nonparametric distribution, namely

---

[2]In practice, this loss is implemented with modifications; the exact form of the training loss is given in Eq. 5.

a Dirichlet Process (DP). Each vector in the set output by the transformer is projected to a pseudo-observation for the DP with distribution $\mathcal{N}(\boldsymbol{\mu_i}, \boldsymbol{\sigma_i^2})$ and pseudo-count $\alpha_i$. In particular, we define $q_\phi(H \mid X)$ as a DP such that [3]:

$$q_\phi(H \mid X) \coloneqq \mathrm{DP}(\mathcal{N}_0(\boldsymbol{\mu}, \boldsymbol{\sigma^2}), \boldsymbol{\alpha_0}) \quad \mathcal{N}_0(\boldsymbol{\mu}, \boldsymbol{\sigma^2}) \coloneqq \sum_{i=1}^n \frac{\boldsymbol{\alpha_i}}{\boldsymbol{\alpha_0}} \mathcal{N}(\boldsymbol{\mu_i}, \boldsymbol{\sigma_i^2}) \quad \boldsymbol{\alpha_0} \coloneqq \sum_{i=1}^n \boldsymbol{\alpha_i} \quad (4)$$

where $\boldsymbol{\mu} \in \mathbb{R}^{n \times p}$, $\boldsymbol{\sigma^2} \in \mathbb{R}^{n \times p}$, $\boldsymbol{\alpha} \in \mathbb{R}^n$ are parameters linearly projected from the $\phi$-parameterized transformer output $V \in \mathbb{R}^{n \times p}$.

As a result, our latent representations are samples from DPs, i.e. mixture distributions whose effective number of components are unbounded and sensitive to input observations[4]. By treating our latent space as a distribution over distributions, we hope to learn a space which is smooth and disentangled, taking inspiration from VAE variants (Burgess et al., 2018). This approach can be viewed as a natural generalization of VAEs from the vector space regime to the set-of-vector space regime, which is appropriate for transformer architectures.

**Abduction Transformer encoder.** Our encoder transformer is trained to approximate $q_\phi(H \mid x_i, y_i)$, where $(x_i, y_i)$ is a pair contained in a problem specification. To give our transformer encoder this probabilistic interpretation, its output is stochastic. The encoder takes a pair from the problem specification $X$, tokenizes it into a set of input vectors, and encodes it into the set of parameters of a DP: $\boldsymbol{\mu}$, $\boldsymbol{\sigma}^2$, and $\boldsymbol{\alpha}$. Then a discrete mixture is sampled from this inferred DP using a factorized sampling method from Henderson & Fehr (2023). The mean of these samples across pairs in $X$ is taken as the encoder's output hypothesis: $\overline{H} = \frac{1}{n} \sum_{i=1}^n H_i$, $H_i \sim q_\phi(H \mid x_i, y_i)$. See App. A for an overview of the operations involved in inferring DP parameters.

**Abduction Transformer decoder.** The decoder specifies the distribution $\hat{y} \sim p_\theta(y_i \mid x_i, H)$. It is implemented as a transformer decoder that autoregressively generates its prediction $\hat{y}$ by cross-attending to the latent representation $H$, and self-attending to the context $x_i$. Since $H$ is represented as a mixture distribution, cross-attention to $H$ involves the denoising-attention operation, which Henderson & Fehr (2023) show theoretically subsumes regular attention.

Thus, because the decoder generates its prediction $y_i$ conditioned on $x_i$ and $H$, it can be viewed as a mechanism which computes $H(x_i)$. In practice, the latent hypothesis $H$ is often the inferred average hypothesis $\overline{H}$ over the pairs in $X$, and $x_i$ is the input query $x_{\text{query}}$. Further details regarding denoising-attention and its equivalence to regular attention can be found in App. B.

**Gradient search over latent hypotheses.** Similarly to Macfarlane & Bonnet (2025), our architecture allows for refinement of latent representations produced by the encoder to further improve congruence with the few-shot examples given in the problem specification. Given some hypothesis $H$, we allow its gradient-based refinement by minimizing $-\sum_{i=1}^n \log p_\theta(y_i \mid x_i, H)$ across few shot examples $\{(x_i, y_i)\}_{i=1}^n$ with respect to $H$.

This process can be thought of as gradient search over latent space to find a hypothesis that better accounts for potentially novel observations, where the search is initialized by a forward pass of the encoder. By decoding each candidate hypothesis into its predictions over $y_i$ at each step, our procedure can be viewed as a form of iterative hypothesis testing and solution verification against few-shot examples. A refined hypothesis $H^o$ is then used to decode some test query $x_{\text{query}}$ in order to predict the ground truth $y^*$, both of which are not seen during the gradient search process.

**Training procedure.** As mentioned in §3, the training loss for these networks is variational free energy, a standard objective used by architectures such as VAEs. We assume a dataset of meta-learning episodes, each containing a problem specification $X$, a test query $x_{\text{query}}$ and ground truth test output $y^*$. For each episode, we compute the loss:

$$\mathcal{L}(\phi, \theta) = \lambda_{\text{KL}} \frac{1}{n} \sum_{i=1}^n \mathrm{KL}(q_\phi(H \mid x_i, y_i) \,\|\, p(H)) - \log p_\theta(y^* \mid x_{\text{query}}, \overline{H}) \quad (5)$$

---

[3]Our implementation incudes an isotropic Gaussian prior component in the DP's base distribution, which is omitted from the expression given here for brevity.

[4]Samples from DPs are theoretically infinite. We truncate samples by only considering $\kappa_0 = n + 1$ components, where $n$ is the number of input vectors with an added prior component.

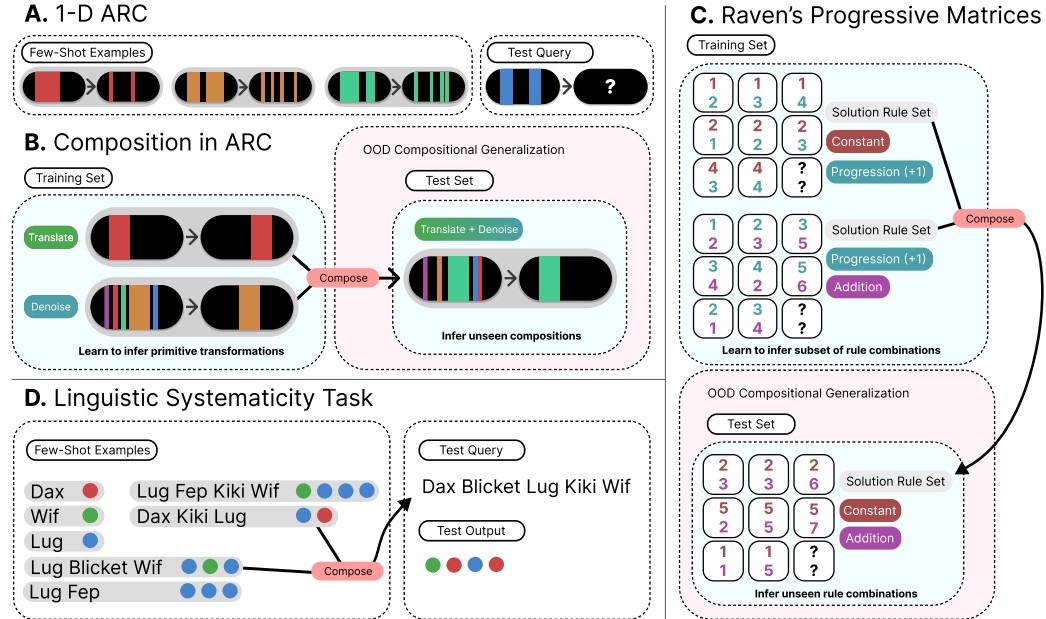

Figure 2: **Illustration of compositional generalization tasks. A** (1-D ARC) A set of 3 few-shot examples are given in the form of input/output pairs, and the objective is to predict the correct output on some test query. **B** (Composition in ARC) We train our model on problems containing a subset of possible transformations. At test time, we evaluate on unseen transformations which are compositions of transformations seen during training. **C** (Raven's Progressive Matrices) Our model is trained on problems containing some subset of possible combinations of rules for feature progressions.[5]At test time, we evaluate on problems with unseen rule combinations. **D** (Linguistic Systematicity Task) Our model trains on meta-learning episodes containing sentence/interpretation pairs. At test-time, the model is evaluated on meta-learning episodes with unseen grammars governing the sentence/interpretation pairs.

and update our parameters $\phi$ and $\theta$ according to back-propagated gradients, where $\lambda_{\text{KL}}$ is a scalar hyperparameter.

The KL-divergence term acts as an information-theoretic regularizer on the posterior DP parameters, encouraging sparsity in the mixture weights given to mixture components (the set of vectors accessible by cross-attention), and encouraging a smoother latent-space due to noisy training-time sampling of component vectors. The prior we use in the KL-divergence term is a DP with an isotropic Gaussian with unit variance as its base distribution, and concentration parameter set to one.

It is also possible to train Abduction Transformer while including intermediate gradient search applied to the average hypothesis $\overline{H}$ before decoding. In this case, the second term in Eq. 5 becomes $-\log p_\theta(y^* \mid x_{\text{query}}, \overline{H}^o)$, where $\overline{H}^o$ is the refined hypothesis after gradient search to optimize the few-shot examples in $X$. Details and pseudo-code for the training procedure, as well as the particular KL-divergence loss we use for DPs, are given in App. C.

# 4 PROGRAM INDUCTION IN 1-D ARC

For our first meta-learning task, we consider 1-D abstract spatial reasoning problems inspired by the ARC-AGI benchmark (Chollet et al., 2025). The task consists of a problem specification $X$ containing input/ouput pairs of pixel sequences which are all governed by some common transformation $H^*$, and the objective is to predict the ground truth output pixel sequence $y^*$ given $x_{\text{query}}$. The task thus contains a perceptual component of discovering entities within a sequence of pixel values, as well as inferring the transformations being applied to them, and is useful as a benchmark for abstract reasoning ability (Chollet, 2019).

Problems of this nature are difficult for neural networks to solve, including architectures incorporating pre-trained LLMs (Xu et al., 2024; Dimitriadis & Samothrakis, 2025; Chollet et al., 2025). In

this section, we seek to investigate the ability of Abduction Transformer to solve ARC-like problems whose solutions are not seen during training, but consist of compositions of transformations seen during training. In other words, we investigate whether Abduction Transformer exhibits compositional generalization with respect to function composition according to the definition given in §2.

**1-D ARC dataset.** For these experiments, we utilize an open-source dataset arc-like (neurallambda, 2024) which allows us to procedurally generate 1-D ARC-like problems with composable combinator functions. We train Abduction Transformer and other baseline architectures on a training set of ARC-like problems with solutions $H \in \mathcal{T}$ with $|\mathcal{T}| = 36$. The set of training hypotheses includes primitive transformations such as translation, color-shift, expansion, sorting, reflection, and denoising, as well as a subset of possible compositions of these primitives. Training examples are generated by sampling a random transformation $H$ from $\mathcal{T}$, generating a random input sequence conditioned on $H$, and then computing the output sequence by applying the predefined combinator functions associated with $H$.

Our training set can be expressed as a set of tuples $D_{\text{train}} = \{(X, x_{\text{query}}, y^*)_j\}_{j=1}^N$ with problem specifications taking the form $X = \{(x_i, y_i)\}_{i=1}^3$. We define $x_i, y_i, x_{\text{query}}, y^*$ as sequences of pixel values such that $H^*(x_i) = y_i$ and $H^*(x_{\text{query}}) = y^*$.

**Experimental setup.** To test OOD compositional generalization, we evaluate our trained models on a test set with solutions $H \in \mathcal{V}$ with $|\mathcal{V}| = 14$ such that $\mathcal{V}$ only contains transformations which are compositions of those found in $\mathcal{T}$, but not themselves found in $\mathcal{T}$. For example, if `translate` and `denoise` are found in $\mathcal{T}$, then $\mathcal{V}$ may contain `translate` ∘ `denoise`. In other words, we design a test set of tuples representing meta-learning episodes $D_{\text{test}} = \{(X, x_{\text{query}}, y^*)_j\}_{j=1}^N$ such that for all $(X, x_{\text{query}}, y^*) \in D_{\text{test}}$, $(X, x_{\text{query}}, y^*)$ is function compositional relative to $\mathcal{T}$. See Fig. 2B for a visual illustration of the training and test sets. The test set contains 2000 such meta-learning episodes which are each generated in the same way as those in the training set. See App. D.1 for a complete enumeration of training and test set transformations.

Table 1: Performance on 1-D ARC OOD Composition Task. Zero-shot test set performance for GPT-5 Thinking and GPT-4.1 (OpenAI, 2024) are provided to contextualize task difficulty (see Appendix. H for details). We report standard error over 3 seeds.

| Model | Solve Rate (%) | Gradient Search Steps | |
|---|---|---|---|
| | | Train | Eval |
| Abduction Transformer **(Ours)** | **25.1** $_{\pm 2.6}$ | 1 | 100 |
| LPN (Macfarlane & Bonnet, 2025) | 1.9 $_{\pm 1.0}$ | 1 | 100 |
| Encoder-decoder Transformer Baseline | 0.1 $_{\pm 0.0}$ | 1 | 100 |
| Decoder-only Transformer Baseline | 5.2 $_{\pm 1.3}$ | None | None |
| *Ablations* | | | |
| Abduction Transformer (No KL-regularization) | 16.7 | 1 | 100 |
| Abduction Transformer (No gradient search) | 0.1 | None | None |
| GPT-5 Thinking (w.o. fine-tuning) | 29.0 | N/A | N/A |
| GPT-4.1 (w.o. fine-tuning) | 11.0 | N/A | N/A |

**Results.** Table 1 compares perfect solve rates across various architectures. We see that Abduction Transformer performs significantly better than any of the baseline architectures, solving 25.1% of problems perfectly. Our results show that gradient search is effective for solving 1D-ARC problems requiring compositional generalization. In addition, we find that using nonparametric latent representations outperforms previous test-time gradient search approaches which utilize single vector latent representations, namely Latent Program Network (Macfarlane & Bonnet, 2025). We also see that our information-theoretic regularization plays an important role, with our ablations on KL-regularization performing worse.

---

[5]Although not illustrated, in order to drastically increase the size of the hypothesis space the feature vector components are randomly permuted column-wise, meaning the row $(1, 2), (1, 3), (1, 4)$ may be modified to $(1, 2), (\mathbf{3}, \mathbf{1}), (1, 4)$ (where feature vectors in the second column of other rows have their components permuted in the same way).

The baseline models with no test-time gradient search both show significantly diminished performance, demonstrating that compositional generalization on 1D-ARC tasks is difficult without test-time adaptation. The effect of scaling the number of gradient search steps is discussed in App. D.3. Finally, we highlight that the deterministic encoder-decoder transformer baseline solves only 0.1% of problems, even when trained with gradient search, indicating that without a stochastic treatment of the encoder the training process does not lead to a searchable latent space. Model and hyperparameter details can be found in App. D.2. Further discussion on the geometry of our learned latent space can be found in App. D.4.

**Verifying non-compositional abilities.** To clarify the role that our nonparametric latent space plays in the improved performance for compositional generalization, we further evaluate both Abduction Transformer and LPN on problems without unseen compositions, i.e., novel instances of problems whose transformations are contained in the training data and thus do not require compositional generalization. Results are presented in App. F. We find that on these problems, both Abduction Transformer and LPN exhibit almost identical performance. This indicates that, in our experiments, the improvements from using a nonparametric latent space are specific to compositional generalization.

## 5    RAVEN'S PROGRESSIVE MATRICES

Our second experiment considers a symbolic variation of Raven's Progressive Matrices, an abstract reasoning task commonly used as a measure of intelligence in humans (Raven, 1962). In its original formulation, a problem instance takes the form of a matrix of panels whose contents evolve row-wise according to some common set of rules, and the objective is to predict the contents of a query panel usually placed on the last column of the last row. The task requires the discovery of a set of rules which explain the progressions seen in the problem, and therefore requires search over possible hypotheses (Carpenter et al., 1990). As such, we frame Raven's Progressive Matrices as a meta-learning problem similar to previous tasks considered, where we take each row of the problem as a few-shot meta-learning example.

**SRAVEN.** We utilize the open source SRAVEN dataset, which is a symbolic variation of Raven's Progressive Matrices specifically developed to test compositional generalization in neural network architectures Schug et al. (2025). Instead of the graphical format which Raven's Progressive Matrices are classically presented in, SRAVEN encodes each panel of a problem instance into a feature vector with feature values from a fixed vocabulary. Unlike the previous ARC-like reasoning task which contains a large perceptual component, SRAVEN is designed to test purely symbolic compositional reasoning, giving us fine-grained control over the rule combinations (i.e. compositions) that we evaluate.

Specifically, SRAVEN encodes each problem panel as a $K$-dimensional vector, where each feature can take on one of $F$-many integer values. Each feature is governed by a progression rule which takes as input two integers and outputs a single integer. Thus, the first two columns of each row in combination with the set of $K$ rules governing each feature determine the right-most panel of each row. See App. E.1 for an overview of the list of progression rules contained in SRAVEN.

Problems are generated by first sampling a set of $K$ rules (out of $N$ total), and producing each row by sampling random inputs to the rules. The final column of the final row is treated as the test query and is hidden to the model. This leads to a $3 \times 3$ matrix $R$ of $K$-dimensional vectors, described as a problem specification $X = \{(x_i, y_i)\}_{i=1}^2$ where $x_i = (R_{i,1}, R_{i,2})$ and $y_i = R_{i,3}$. In addition, $x_{\text{query}}$ is defined as $(R_{3,1}, R_{3,2})$ and $y^*$ as $R_{3,3}$. To increase the difficulty of our problems, the components of feature vectors are randomly permuted in the same way for each column in the problem across few-shot examples. Identically to the previous experiments, training sets take the form of a set of tuples $D_{\text{train}} = \{(X, x_{\text{query}}, y^*)_j\}_{j=1}^N$ and the same format applies to our test sets.

**Experimental setup.** In order to evaluate OOD compositional generalization ability, we generate two datasets with $N = 8$, $K = 4$ and $F = 8$ such that the rule combinations seen in problems of each dataset are disjoint.[6] For instance, if the training set contains a problem instance with rule set $\{A, B\}$, then the test does not contain any problem with that rule set. See Fig. 2C for an illustration of our dataset split. Out of all possible combinations of $K$ rules, we randomly sample

---

[6]This gives $\binom{R+K-1}{K} \cdot (K!^2 - K) = 188,760$ many possible SRAVEN tasks (possible hypotheses) in total, taking feature permutations into account.

a fraction of these for our training set; for this experiment, we train models on 1% of possible rule combinations, as well as on 90% of possible rule combinations, and evaluate on problems containing the held out proportion of rule combinations. In our few-shot meta-learning framework, this amounts to performing evaluations on test sets whose problems are function compositional relative to the training set. Our test sets consist of 2000 generated OOD SRAVEN problems each.

Table 2: SRAVEN OOD Composition Task: Performance when training on 1% and 90% of possible rule combinations.[7] For each training method, we evaluate on problems with rule combinations taken from the remaining unseen proportion of combinations. We additionally present zero-shot performance on the test set containing 99% of rule combinations for GPT-5 Thinking and GPT-4.1.

| Model | Train on 1% | | | Train on 90% | | |
|---|---|---|---|---|---|---|
| | Solve Rate (%) | Gradient Steps | | Solve Rate (%) | Gradient Steps | |
| | | Train | Eval | | Train | Eval |
| Abduction Transformer (**Ours**) | **46.1** $_{\pm 4.2}$ | 1 | 100 | 96.4$_{\pm 0.4}$ | 1 | 10 |
| LPN (Macfarlane & Bonnet, 2025) | 37.1 $_{\pm 2.0}$ | 1 | 100 | 93.5$_{\pm 1.0}$ | 1 | 10 |
| Encoder–decoder Transformer Baseline | 10.8 $_{\pm 2.2}$ | 1 | 100 | 27.2$_{\pm 3.0}$ | 1 | 10 |
| Decoder-only Transformer Baseline | 28.8 $_{\pm 1.3}$ | None | None | 95.3 $_{\pm 1.1}$ | None | None |
| *Ablations* | | | | | | |
| Abduction Transformer (No KL-regularization) | 16.8 | 1 | 100 | 33.7 | 1 | 10 |
| Abduction Transformer (No gradient search) | 20.9 | None | None | 96.7 | None | None |
| GPT-5 Thinking (w.o. fine-tuning) | 41.0 | N/A | N/A | N/A | N/A | N/A |
| GPT-4.1 (w.o. fine-tuning) | 34.0 | N/A | N/A | N/A | N/A | N/A |

**Results.** Our results are summarized in Table 2. We find that when trained on 90% of possible compositions, Abduction Transformer and the decoder-only transformer baseline perform comparably, showing close to perfect performance on the OOD test set. When trained on 1% of possible compositions, we find that the Abduction Transformer significantly outperforms all other architectures we tested. We observe that the decoder-only baseline which performed well in the previous setting now degrades significantly in performance, indicating poor generalization in more extreme OOD regimes. The LPN similarly equipped with test time gradient search performs better than the decoder-only baseline but not as well as Abduction Transformer, indicating the advantage of using nonparametric latent representations.

Our ablations show that 1) KL-regularization during training is necessary for effective latent space search, 2) gradient search is needed for more extreme OOD compositional generalization and 3) gradient search fails when applied to a deterministic encoder-decoder architecture, indicating that our probabilistic treatment is necessary. Model and hyperparamter details can be found in App. E.2. We discuss the effect of scaling gradient search steps in App. D.3.

**Verifying non-compositional abilities.** Similarly to our previous experiments, we perform additional evaluations on non-compositional SRAVEN problems. We again find that Abduction Transformer and LPN perform almost identically in this regime. Results are presented in App. F.

## 6 LINGUISTIC SYSTEMATICITY

Finally, we consider a grammar induction task proposed by Lake & Baroni (2023) to test linguistic systematicity and compositional generalization ability in neural networks. The task is a meta-learning task where each meta-learning episode contains 14 few-shot examples of sentence/interpretations pairs, namely pairs consisting of a sequence of lexical symbols and a sequence of colored circles. The objective of the task is to infer the underlying interpretation grammar from these examples and ascertain the correct interpretation of some query sentence. See Fig. 2D for an illustration of the problem setup.

**Experimental setup.** We use the open source training and test sets created by Lake & Baroni (2023) to train and evaluate our models. The training set is a set of meta-learning episodes $D_{\text{train}} = \{(X, x_{\text{query}}, y^*)_j\}_{j=1}^{N}$ each generated from a randomly sampled interpretation grammar, with $X = \{(x_i, y_i)\}_{i=1}^{14}$ consisting of pairs containing a lexical symbol sequence $x_i$ and a color sequence $y_i$. The pairs in $X$ are generated such that given the ground truth interpretation grammar $H^*$, we have

---

[7] Reported standard error is computed over 3 seeds.

that $H^*(x_i) = y_i$. The test set is generated in the same way from interpretation grammars that are guaranteed to be different from those used in the training set. Thus, in this experiment we evaluate our models' ability to perform production compositional generalization. See App. G for details on the sampling process for interpretation grammars.

We compare the performance of Abduction Transformer against the encoder-decoder transformer architecture from Lake & Baroni (2023) as a baseline while decreasing the number of few-shot examples in test problems. With this setup, we aim to evaluate OOD compositional generalization ability as few-shot examples become scarce to the point where they may not be sufficient to imply a unique interpretation grammar. In our experiments, we vary the number of few-shot examples given at test-time in the range $[1, 14]$ and evaluate how well our models generalize to these situations.

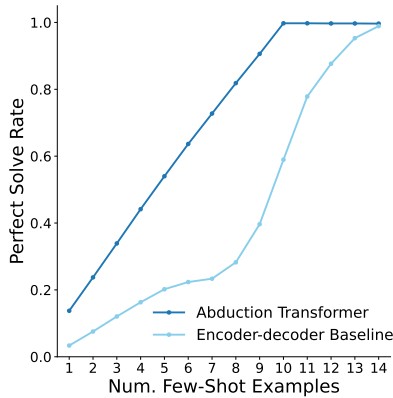

Figure 3: Performance on linguistic systematicity task when varying the number of few-shot examples given in meta-learning episodes at test time.

**Results.** As shown in Fig. 3, we find that compared to the encoder-decoder baseline, Abduction Transformer exhibits near perfect accuracy down to 10 few-shot examples before degrading, maintaining 50% accuracy at 5 few-shot examples. This is in contrast to the baseline architecture which shows continuous degradation in performance as the number of few-shot examples decreases. This indicates Abduction Transformer's strong compositional generalization ability in the face of incomplete information, considering the fact that the ground truth interpretation grammars in the test set contain exactly 7 rewrite rules and require the observation of at least 7 examples to be uniquely identified. Our results highlight Abduction Transformer's robustness against OOD regimes, both in terms of the meta-learning task itself and the scarcity of few-shot examples.

## 7 RELATED WORK

**Test-time adaptation methods.** Performing gradient updates at test-time has become a popular approach in order to solve difficult inference problems. A common approach is Test-Time-Fine-Tuning (TTFT), a method for improving inference performance by fine-tuning models on some curated dataset conditioned on the test-input (Hübotter et al., 2025; Krause et al., 2018; Hardt & Sun, 2024; Sun et al., 2020). Similar approaches have been used in successful architectures for the ARC-AGI benchmark (Chollet et al., 2025; Akyürek et al., 2025; Franzen et al., 2025). Approaches using updates over model activations enable test-time adaptive performance increases while avoiding model parameter updates (Macfarlane & Bonnet, 2025; Li et al., 2025).

**Compositional generalization.** The ability for neural networks to solve OOD compositional reasoning problems has been well studied in the context of LLMs (Kudo et al., 2023; An et al., 2023; Liu et al., 2023; Mészáros et al., 2024). Hosseini et al. (2022); Furrer et al. (2021) indicate the presence of compositional generalization ability in LLMs for parsing tasks, and Schug et al. (2025) investigate the mechanistic origins of compositional generalization in transformer architectures by framing attention as a hypernetwork. Despite this, many studies have shown that LLMs, including those trained specifically for reasoning, fail on OOD problems that require compositional recombination of learned knowledge and subroutines (Dziri et al., 2023; Mirzadeh et al., 2025; Opedal et al., 2025; Shojaee et al., 2025). Thus, the problem of whether transformer architectures (and neural networks in general) can learn to compositionally generalize remains open.

## 8 CONCLUSION

We introduced Abduction Transformer as a novel architecture capable of OOD compositional generalization on reasoning tasks, using nonparametric mixture distribution latent representations, information-theoretic regularization, and test-time gradient search. Our architecture learns to infer initial distributions over mixture distributions in a smooth space of hypotheses which then supports the iterative refinement of these hypotheses at test-time to better account for few-shot meta-learning

examples, thereby solving problems involving unseen compositions of knowledge obtained during training.

Our experiments show that Abduction Transformer is capable of solving problems involving such unseen compositions in ARC-like reasoning tasks, symbolic Raven's Progressive Matrices, and grammar induction tasks, even in extreme OOD settings, where standard transformer architectures and previous test-time latent space search methods fail. We show that performance drops significantly without our test-time gradient search procedure, and that the search process becomes less effective without information-theoretic regularization, nonparametric representations, or stochastic training. Our method serves as a new direction in designing neural network architectures that are capable of complex OOD generalization in reasoning domains.

ACKNOWLEDGMENTS

We thank Fabio Fehr, Andrei Coman, Philip Garner, and Itsaso Olasagasti for helpful discussion and feedback. This work was funded in part by the Swiss National Science Foundation under the NCCR grant Evolving Language, Swiss National Science Foundation Agreement #51NF40_225146.

**Ethics statement.** This work is largely foundational, demonstrating the viability of a new architectural paradigm. We foresee no harmful applications of our methods, nor potential for discrimination, bias or fairness concerns. No LLMs were used in the writing or proofreading of the main text. See App. I for details on LLM use.

**Reproducibility statement.** Code will be published at `https://github.com/idiap/AbductionTransformer`. We provide information to reproduce our experiments in App. D and App. E.

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

## A  ABDUCTION TRANSFORMER ENCODER

Given transformer outputs $V \in \mathbb{R}^{n \times p}$, the encoder linearly projects these representations to pre-activtion parameters $\boldsymbol{\mu} \in \mathbb{R}^{n \times p}$, $\log(\boldsymbol{\sigma^2}) \in \mathbb{R}^{n \times p}$, $\boldsymbol{\alpha} \in \mathbb{R}^n$. Following Henderson & Fehr (2023), we apply an exponential activation function to our log-variance parameters and ReLU (Nair & Hinton, 2010) to our concentration parameters. This gives a posterior mixture distribution containing one component $\langle \boldsymbol{\mu}_i, \boldsymbol{\sigma}_i, \alpha_i \rangle$ per transformer output, along with the prior component.

## B    DENOISING ATTENTION

In standard attention, a query vector accesses a set of vectors through a weighted sum parameterized by its attention vector. In denoising attention (Henderson & Fehr, 2023), this interpretation is extended by generalizing sets of vectors to probability distributions over vectors, and treating attention as a function of these probability distributions.

**Scaled dot-product attention.** We first consider standard cross-attention where some input query is mapped so a single result vector. Here, the input $u' \in \mathbb{R}^{1 \times p}$ is projected to the query through $W^Q \in \mathbb{R}^{p \times d}$. We obtain keys and values by projecting the set of vectors $Z \in \mathbb{R}^{n \times p}$ through $W^K, W^V \in \mathbb{R}^{p \times d}$ respectively. We can regroup the standard expression of the attention function to operate in the space of $Z$, that is:

$$\text{Attention}(u', Z; W^Q, W^K, W^V) = \text{Attn}(u' W^Q (W^K)^\top, Z) W^V = \text{Attn}(u, Z) W^V \quad (6)$$

where $u = u' W^Q (W^K)^\top, u \in \mathbb{R}^{1 \times p}$. This new operation $\text{Attn}(u, Z)$ can be characterized both as a sum over vectors $z_i \in Z$, or in terms of an integral over a distribution with support at $z_i$. This equivalence is restated below:

$$\text{Attn}(u, Z) = \text{softmax}\left(\frac{1}{\sqrt{d}} u Z^T\right) Z = \text{DAttn}(u; F_Z) \quad (7)$$

$$F_Z = \sum_{i=1}^n \frac{\exp\left(\frac{1}{2\sqrt{d}}\|z_i\|^2\right)}{\sum_{i=1}^n \exp\left(\frac{1}{2\sqrt{d}}\|z_i\|^2\right)} \delta_{z_i} \quad (8)$$

$$\text{DAttn}(u; F) = \frac{\int_v f(v)\, g(u; v, \sqrt{d}I)\, v\, dv}{\int_v f(v)\, g(u; v, \sqrt{d}I)\, dv} \quad (9)$$

Here, $\delta_{z_i}$ is an impulse distribution (Dirac delta function) at $z_i$, $f(\cdot)$ is the probability density function for distribution $F$, and $g(u; , v, \sqrt{d}I)$ is a Gaussian distribution with diagonal covariance.

**Interpretation as denoising.** The operation $\text{DAttn}(u; F_Z)$ which we call *denoising attention* can be thought of as the mean of the posterior distribution (over $v$) induced by making an observation $u$ of some true vector $v$ corrupted by noise, where $v$ is drawn from a prior distribution $F_Z$ specified by $Z$. Denoising attention is a generalization of regular attention, where instead of restricting to sets of vectors, it is defined for any distribution $F$ over a vector space. When the input distribution $F$ is discrete, it can be implemented naturally by including a bias term in the cross-attention operation.

## C    TRAINING DETAILS: PSEUDO CODE AND KL-DIVERGENCE

**Pseudo code.** 1 shows our training procedure with gradient search enabled.

---

**Algorithm 1:** Abduction Transformer training procedure with gradient search

---

**Input:** Encoder parameters $\phi$, decoder parameters $\theta$

1 **for** $t \leftarrow 1$ **to** `num_training_steps` **do**

2     Draw problem specification with $n$ input–output pairs $(x_i, y_i)$, test query $x_{\text{query}}$ and ground truth test output $y^*$

    /* Sampling from DP                                        */

3     **for** $i \leftarrow 1$ **to** $n$ **do**

4        Sample $H_i \sim q_\phi(H \mid x_i, y_i)$

    /* Gradient search                                          */

5     $\overline{H} \leftarrow \frac{1}{n} \sum_{i=1}^{n} H_i$

6     **for** $k \leftarrow 1$ **to** `num_gradient_search_steps` **do**

7        $\overline{H} \leftarrow \overline{H} + \mu \cdot \nabla_H \sum_{i=1}^{n} \log p_\theta(y_j \mid x_j, H) \Big|_{H=\overline{H}}$

       // We ignore the second-order gradient w.r.t. $\theta$

8     $H^o \leftarrow \overline{H}$

9     $\mathcal{L}(\phi, \theta) = \frac{1}{n} \sum_{i=1}^{n} \lambda_{\text{KL}} \text{KL}(q_\phi(H \mid x_i, y_i) \,\|\, p(H)) - \log p_\theta(y^* \mid x_{\text{query}}, H^o)$

10     Update $\phi$ and $\theta$ via gradient descent on $\mathcal{L}(\phi, \theta)$

---

**KL-divergence for Dirichlet processes.** To compute the KL-divergence between DPs, we use an approximation due to Henderson & Fehr (2023). Here, superscripts $q$ and $p$ denote that the particular DP parameter belongs to the posterior and prior DP, respectively. Note that $\kappa_0 = n + 1$, where $n$ is the number of input vectors (the additional term accounts for the prior component). $\Gamma$ and $\psi$ refer to the gamma function and digamma function, respectively.

$$D_{\text{KL}}\big(q(H \mid X) \,\|\, p(H)\big) \approx L_D + L_G \tag{10}$$

$$L_D = \log \Gamma(\alpha_0^q) - \log \Gamma(\alpha_0^p) + (\alpha_0^q - \alpha_0^p) \left( -\psi(\alpha_0^q) + \psi\left(\tfrac{\alpha_0^q}{\kappa_0}\right) \right)$$
$$+ \kappa_0 \left( \log \Gamma\left(\tfrac{\alpha_0^p}{\kappa_0}\right) - \log \Gamma\left(\tfrac{\alpha_0^q}{\kappa_0}\right) \right) \tag{11}$$

$$L_G = \tfrac{1}{2} \kappa_0 \sum_{i=1}^{n+1} \frac{\alpha_i^q}{\alpha_0^q} \sum_{h=1}^{d} \left[ \frac{(\mu_{ih}^q - \mu_h^p)^2}{(\sigma_h^p)^2} + \frac{(\sigma_{ih}^q)^2}{(\sigma_h^p)^2} - 1 - \log \frac{(\sigma_{ih}^q)^2}{(\sigma_h^p)^2} \right] \tag{12}$$

# D    1-D ARC

## D.1    1-D ARC TRANSFORMATIONS

We show the transformations contained in the training set and test set in Figures 4 and 5, respectively. Each pixel sequence is plotted as a horizontal array. We plot 3 randomly generated input/output pairs for each transformation. The labels ('E.g.') on the y-axis mark each input/output pair.

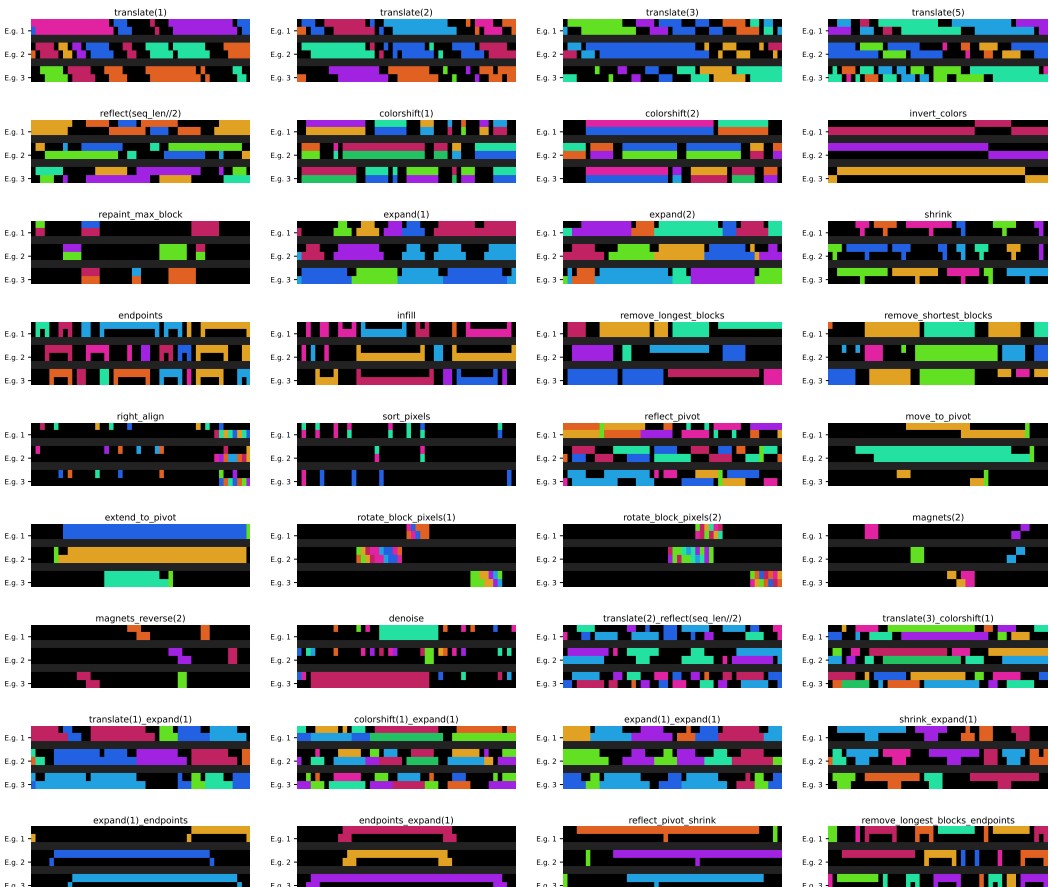

Figure 4: **1-D ARC Training set transformations**

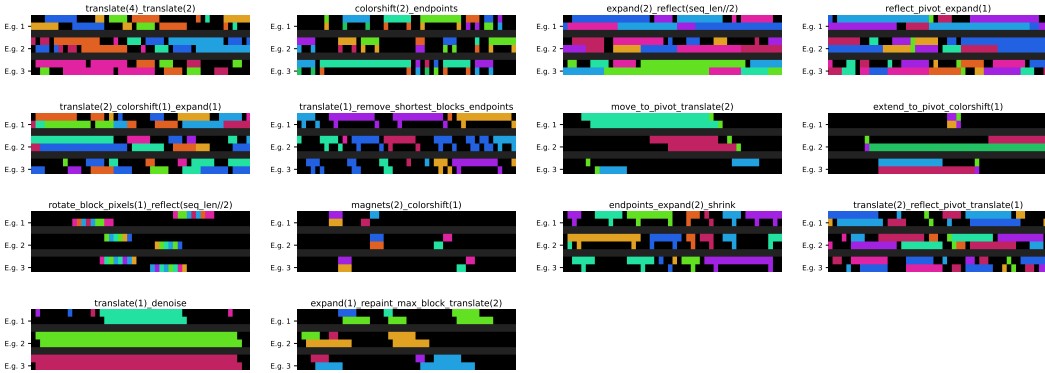

Figure 5: **1-D ARC Test set transformations**

## D.2 ARC-LIKE TASK MODEL DETAILS

Model hyperparamters are shown in Table 3. Training parameters used across all architectures are shown in Table 4.

**Transformer layers.** All of the models we test utilize transformer layers that are structured in the usual way:

$$x_{\text{attn}} = \text{MHA}(\text{LN}(x_{\text{in}})) + x_{\text{in}}$$
$$x_{\text{out}} = \text{FFN}(\text{LN}(x_{\text{attn}})) + x_{\text{attn}}$$

Our FFN is a standard MLP with a single hidden layer, using SiLU (Elfwing et al., 2018) activation. We use learned positional encodings and one-hot encode integer inputs corresponding to pixel/symbol values. Both positional and token embeddings are accessed via a dense embedding matrix.

**Generation method.** For all the models we test, we generate predictions autoregressively. For Abduction Transformer, LPN and the encoder-decoder baseline, a transformer decoder autoregressively generates its predictions by self-attending to the test input and cross-attending to the encoded context. For the decoder-only baseline, the test input is given as a prefix, and the prediction is generated by direct autoregression conditioned on the prefix.

Table 3: Details for ARC models.

|                     | Abduction Transformer | LPN       | Encoder-decoder Baseline | Decoder-only Baseline |
|---------------------|-----------------------|-----------|--------------------------|-----------------------|
| Num. Parameters     | 1,255,393             | 1,273,152 | 1,255,393                | 1,389,120             |
| Encoder Layers      | 4                     | 4         | 4                        | 0                     |
| Decoder Layers      | 5                     | 7         | 5                        | 12                    |
| Emb. Dim.           | 96                    | 96        | 96                       | 96                    |
| MLP Dim.            | 384                   | 384       | 384                      | 384                   |
| No. Heads           | 6                     | 6         | 6                        | 6                     |
| Dirichlet KL Coef.  | 0.1                   | N/A       | N/A                      | N/A                   |
| Gaussian KL Coef.   | 0.001                 | 0.001     | N/A                      | N/A                   |
| Gradient Search LR  | 0.1                   | 0.1       | N/A                      | N/A                   |

Table 4: Training hyperparameters. The decoder-only baseline differs only in the batch size used for training (shown in parentheses). All models converged in training loss and validation accuracy before reaching max. training steps.

| Hyperparameter         | Value     |
|------------------------|-----------|
| Training Steps         | 30,000    |
| Batch Size             | 1024 (32) |
| Optimizer              | AdamW     |
| Gradient Clipping Norm | 1.0       |
| Learning Rate          | 0.001     |

### D.3    SCALING GRADIENT SEARCH STEPS

Fig. 6 shows the effect of scaling the number of gradient search steps we take during test-time search. Both Abduction Transformer and LPN benefit from an increased number of search steps on both tasks.

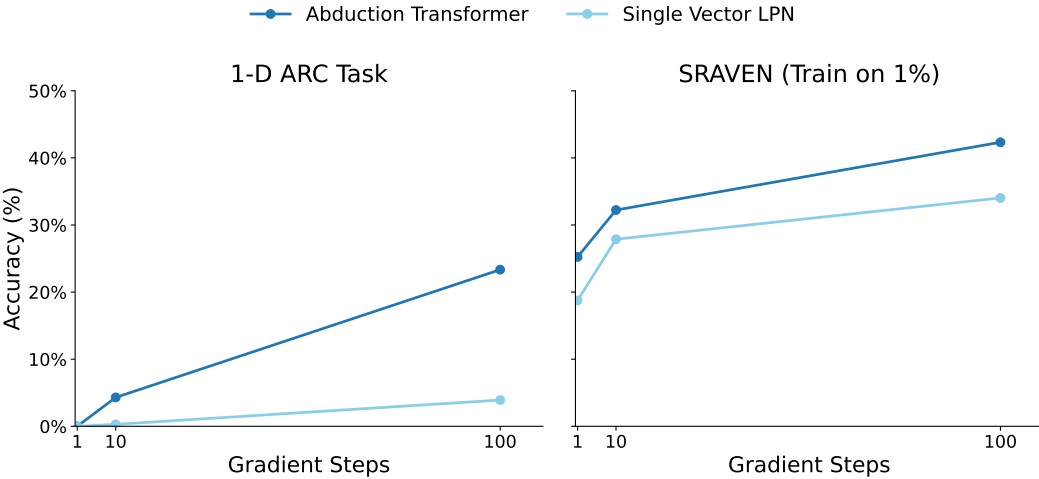

Figure 6: **Accuracy against number of gradient search steps at test-time. Left:** Performance of Abduction Transformer and LPN on the 1-D ARC OOD composition task. **Right:** Performance of the same models on SRAVEN OOD composition task where we train on 1% of rule combinations.

## D.4 ABDUCTION TRANSFORMER LATENT SPACE

**Visualizing latent space.** We hypothesize that a latent space amenable to compositional generalization requires the representations of various hypotheses to be well organized in its geometry according to their semantics. To understand Abduction Transformer's latent space, we sample 15,360 randomly generated 1-D ARC problems from our training and test distributions and plot their representations[8]. Our mixture distribution representations are first flattened into single vectors by taking their expectation, then projected into 2D using t-SNE; the results are presented in Fig. 7. Our plots demonstrate that Abduction Transformer's latent space is remarkably well separated across different primitive transformations seen during training. Furthermore, the representations of unseen compositions map onto sensible locations near their constituent transformations.

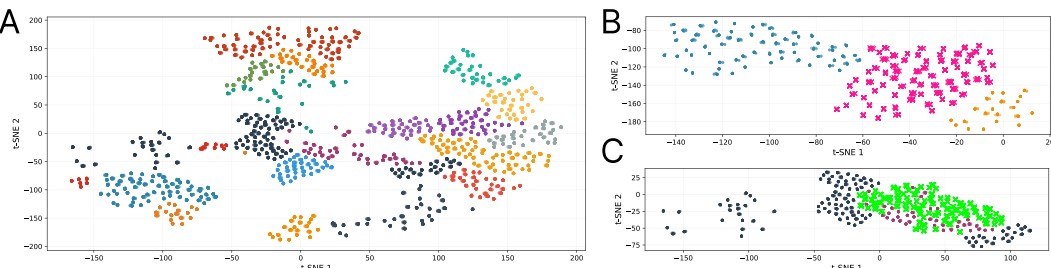

Figure 7: **Visualization of latent space A** (Primitive Transformations) t-SNE plot showing the structure of Abduction Transformer's latent space. Each color corresponds to a particular primitive 1-D ARC transformation. Our mixture distribution representations are flattened by taking their expectation. **B, C** (Unseen Compositions) Latent representations of unseen 1-D ARC problem specifications projected to the same space. **B** shows the representations for `colorshift` in blue, `endpoints` in orange and the unseen composition `colorshift ∘ endpoints` in magenta. **C** shows `translate` in black, `denoise` in purple, and the unseen composition `translate ∘ denoise` in green.

---

[8]We encode problem specifications using an Abduction Transformer instance trained with 1 step of gradient search on the training set. The reported representations are taken from the initial encoding before test-time gradient search is applied.

# E  SRAVEN

## E.1  SRAVEN PROGRESSION RULES

List of SRAVEN progression rules, taken from Schug et al. (2025). $F$ refers to the size of the feature vocabulary.

1. **Constant:** Each row consists of a random but fixed integer from $\{1, \ldots, F\}$.

2. **Progression** (+1)**:** The first element of each row is sampled uniformly at random and incremented by 1 modulo $F$ for each successive column.

3. **Progression** (+2)**:** The first element of each row is sampled uniformly at random and incremented by 2 modulo $F$ for each successive column.

4. **Progression** (−1)**:** The first entry is sampled randomly, and each following entry is decremented by 1 modulo $F$.

5. **Progression** (−2)**:** The first element of each row is sampled uniformly at random and decremented by 2 modulo $F$ for each successive column.

6. **Addition:** Two elements are sampled uniformly at random for each row and added modulo F to obtain the last column.

7. **Subtraction:** Two elements are sampled uniformly at random for each row and subtracted modulo F to obtain the last column.

8. **Distribute three:** Three elements are sampled uniformly at random and presented in three independently sampled random permutations for each row.

## E.2  SRAVEN MODEL DETAILS

Model hyperparamters are shown in Table 5. Training parameters are identical to those shown in Table 4. Details regarding the transformer layers and generation method used in all models are identical to those found in §D.2.

Table 5: Details for SRAVEN models.

|  | Abduction Transformer | LPN | Encoder-decoder Baseline | Decoder-only Baseline |
|---|---|---|---|---|
| Num. Parameters | 1,093,441 | 1,148,640 | 1,093,441 | 1,131,840 |
| Encoder Layers | 4 | 4 | 4 | 0 |
| Decoder Layers | 4 | 6 | 4 | 10 |
| Emb. Dim. | 96 | 96 | 96 | 96 |
| MLP Dim. | 384 | 384 | 384 | 384 |
| No. Heads | 6 | 6 | 6 | 6 |
| Dirichlet KL Coef. | 0.1 | N/A | N/A | N/A |
| Gaussian KL Coef. | 0.001 | 0.001 | N/A | N/A |
| Gradient Search LR | 0.1 | 0.1 | N/A | N/A |

# F  VERIFYING NON-COMPOSITIONAL ABILITIES

Table 6: Performance on non-compositional 1-D ARC problems.

| Model | Solve Rate (%) | Gradient Search Steps | |
|---|---|---|---|
|  |  | Train | Eval |
| Abduction Transformer (**Ours**) | 98.39 | 1 | 100 |
| LPN | 97.90 | 1 | 100 |

Table 7: Performance on non-compositional SRAVEN problems.

| Model | Solve Rate (%) | Gradient Search Steps | |
|---|---|---|---|
| | | Train | Eval |
| Abduction Transformer (**Ours**) | 99.95 | 1 | 100 |
| LPN | 99.90 | 1 | 100 |

## G  INTERPRETATION GRAMMAR AND META-LEARNING INSTANCE SAMPLING

An example interpretation grammar, taken from Lake & Baroni (2023) is shown in Fig. 8. Interpretation grammars (which each correspond to individual meta-learning instances) are randomly generated from a simple meta-grammar.

Rewrite rules for primitives (the first 4 rules in Fig. 8) are generated by randomly sampling input and output symbol pairs without replacement. Rewrite rules for functions are generated by first sampling the LHS, followed by the RHS. The LHS is generated by sampling (without replacement) an input symbol at random, then sampling whether the function is unary/binary, then sampling either primitve or non-primitive variables as its arguments ($u$ and $x$ respectively in Fig. 8). The RHS is generated by sampling a random string of length $\leq 8$ consisting of any of the function arguments defined in the LHS.

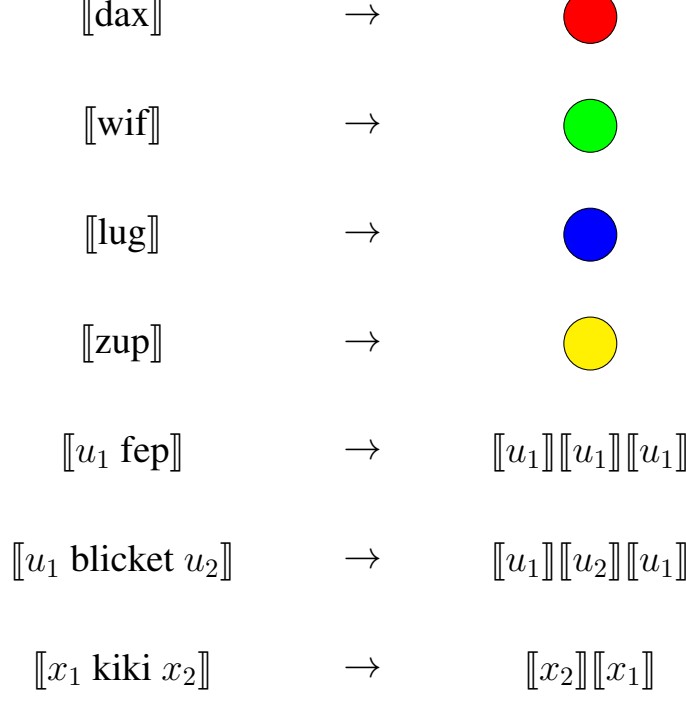

Figure 8: **Example interpretation grammar.** Double brackets $[\![\cdot]\!]$ denote the interpretation function. Variables $x_i$ apply to arbitrary non-empty strings, while $u_i$ apply only to *dax*, *wif*, *lug*, and *zup*.

## H  EXPERIMENTS ON GPT MODELS

We use prompts shown in Fig. 9, and Fig. 10 for our 1-D ARC and SRAVEN experiments on GPT-5 Thinking and GPT-4.1. We allow a maximum of $10,000$ reasoning tokens for GPT-5 Thinking (which corresponds to 'effort' set to 'low' for our tasks), and a maximum of $10,000$ output tokens for GPT-4.1. We evaluate on $400$ randomly selected problems from the test set for 1-D ARC, and $200$ for SRAVEN.

```
1-D ARC Prompt

You are solving a 1D pattern recognition puzzle.  Each puzzle
consists of sequences of pixel values from 0-9, where 0 represents
empty/background.

You will be shown several input-output examples that demonstrate a
transformation rule.  Your task is to identify the pattern and apply
it to a new test input.

EXAMPLES:
Example 1:
Input:  <sequence>
Output:  <sequence>

...

TEST:
Input:  <sequence>
Output:  ?

You may think through the problem in detail, but make sure to end
your response with the final answer in this exact format:

FINAL ANSWER: [your sequence here]

The sequence should be space-separated integers only.
```

Figure 9: Prompt used for 1-D ARC problems.

---

**SRAVEN Prompt**

```
You are solving a SRAVEN (Symbolic RAven's) puzzle.  This is a
visual reasoning task adapted to sequences.

Each puzzle consists of a grid of visual panels, where each panel
is represented by a sequence of feature values (integers 0-7).  Each
row in the grid follows a consistent rule or pattern across its
panels.

You will be shown several rows as examples, where each row contains:
- Input:  The first few panels of the row (showing the pattern)
- Output:  The final panel that completes the pattern

Your task is to identify the underlying rule and apply it to predict
the missing final panel in the test row.

EXAMPLES:

EXAMPLES:
Example 1:
Input:  <sequence>
Output:  <sequence>

...

TEST:
Input:  <sequence>
Output:  ?

Analyze the pattern across the example rows and apply the same rule
to complete the test row.

You may think through the problem step by step, but make sure to end
your response with the final answer in this exact format:

FINAL ANSWER: [your panel here]

The panel should be a comma-separated list of integers, e.g., [1, 2,
3, 4]
```

Figure 10: Prompt used for SRAVEN problems.

## I USE OF LARGE LANGUAGE MODELS

No LLMs were used in the writing or proofreading of the main text. We use LLM assisted tools for writing code used in models and experiments. We use LLM assistance for formatting LaTex figures.

