# OpenReview forum: "Compositional Generalization through Gradient Search in Nonparametric Latent Space"
_ICLR.cc/2026/Conference — ICLR 2026 Poster_

### Official Review · Reviewer_VaZe · 2025-10-26

**Soundness:** 2
**Presentation:** 2
**Contribution:** 1
**Rating:** 2
**Confidence:** 2

**Summary:**

This paper addresses the problem of compositional generalization and proposes the 'Abduction Transformer' as a solution. The model's core mechanism involves inferring Dirichlet Process Posterior parameters to effectively handle few-shot problems. The proposed algorithm consists of a 2-stage process: 1) An Encoder generates an initial hypothesis based on all few-shot examples, and 2) A separate Decoder architecture receives this hypothesis and performs gradient search (fine-tuning) to refine it into a hypothesis that best explains the examples. The authors validate their methodology on benchmarks such as ARC-LIKE REASONING and RAVEN’S PROGRESSIVE MATRICES.

**Strengths:**

This few-shot learning framework is generalizable, so can be applied to any benchmark.

**Weaknesses:**

This paper suffers from critical flaws in its experimental validation, primarily concerning benchmark and baseline selection.

1. Omission of Core Baselines (MAML): The authors acknowledge that their methodology is a ""method of fine-tuning to minimize loss for a given few-shot input,"" and is fundamentally a meta-learning (learn-to-learn) approach. Therefore, the validity of this methodology must be directly compared against at least classic meta-learning algorithms like MAML. They should have verified whether their 'gradient-based hypothesis refinement' mechanism is superior to MAML's 'gradient-based model adaptation' on standard few-shot benchmarks (e.g., tieredImageNet). If their methodology indeed has such generalization capabilities, it must at least outperform classic meta-learning methods. Their method is, in fact, generally applicable. However, the paper intentionally avoids this crucial comparison, presenting only 'strawman' baselines like GPT and standard Transformers, which were not designed for meta-learning. This fails to prove any practical comparative advantage.

2. Use of Arbitrary Benchmarks: The authors used a niche, self-created benchmark called ""1-D ARC-LIKE REASONING"", which they only state is ""inspired by"" the standard 2D ARC benchmark. A search for this benchmark reveals it is extremely niche. Furthermore, even the standard 2D ARC benchmark is not frequently used. On the contrary, on the 2D ARC benchmark—which should be more difficult—some existing LLMs have achieved nearly 70% accuracy. How does this discrepancy arise? Moreover, seeing that modern LLMs like GROK achieve nearly 0% on that benchmark suggests the benchmark has high variance between LLMs and is a task that LLMs are inherently bad at. Why does high performance here validate this model? For example, VLMs perform poorly at depth estimation. A very simple depth estimation model will easily outperform a VLM. What value does that simple model have? This merely proves the VLM's inability on that task; it does not guarantee the new model's performance. This appears to be an attempt to avoid direct comparison on recognized benchmarks, severely damaging the credibility and generalizability of the experimental results.

3. Unrealistic Inference Cost (Test-Time Training): The most significant problem is that the method performs 100 steps of gradient search (fine-tuning) to solve a single test query. This is not fast few-shot inference. This is, in effect, Test-Time Training. This approach of training a model for 100 steps on a single problem renders any comparison against MAML (which takes only a few steps) or GPT (which requires no extra training) fundamentally meaningless. This is an unrealistic inference cost and not a fair comparison."

**Questions:**

See weakness

---

> ### Author Response · Authors · 2025-11-15
>
> Thank you for your detailed review and feedback. We respond to your questions and reservations below.
>
> > The authors acknowledge that their methodology is a ""method of fine-tuning to minimize loss for a given few-shot input,""
>
> We would like to clarify that the quotation cited by the reviewer was never mentioned in our paper (we presume it is the reviewer’s interpretation of our gradient search over latent representations). We stress that our method differs from fine-tuning methods which involve model parameter updates, as we only update a set of latent vectors of length at most the input sequence length of the few-shot examples.
>
> > Therefore, the validity of this methodology must be directly compared against at least classic meta-learning algorithms like MAML.
>
> Thank you for raising this point, but we are contributing to a different topic within meta-learning. Although MAML is a classic method in meta-learning for adapting model weights at test-time, this paper chooses to highlight the effectiveness of simply updating model activations and avoiding model updates entirely, and compare against baselines which also update their latent representations with frozen model weights. By performing search over latent representations as opposed to model parameters, we update far fewer parameters: 10^3 parameters for latent space search vs 10^6 parameters for full model parameter updates. We have made this clearer in an updated version of the main text.
>
> > However, the paper intentionally avoids this crucial comparison, presenting only 'strawman' baselines like GPT and standard Transformers, which were not designed for meta-learning.
>
> The state of the art in this area is GPT-style models and other standard Transformers.  These LLMs have given rise to an important breakthrough in meta-learning abilities through the phenomenon of in-context learning. Evidence of this has been demonstrated overabundantly in the literature, but we cite some influential works here [1] [2] [3] [4] [5].
> While we cannot perform controlled experiments comparing to these large pretrained models, we provide results with a model in this same class so as to contextualize our results.  This comparison clearly shows that our results are competitive with these state-of-the-art models, thereby showing that the positive results in our controlled experiments are not simply due to comparing to strawman baselines.  In particular, we show that the domain of compositional generalization is a particularly challenging task even for these powerful transformers, and contribute a method for combining the power of transformers’ attention-based representations with test-time search over these representations.
>
> [1] Brown, T., Mann, B., Ryder, N., Subbiah, M., Kaplan, J. D., Dhariwal, P., ... & Amodei, D. (2020). Language models are few-shot learners.
>
> [2] Wei, J., Bosma, M., Zhao, V. Y., Guu, K., Yu, A. W., Lester, B., Du, N., Dai, A. M., and Le, Q. V. (2022). Finetuned Language Models Are Zero-Shot Learners.
>
> [3] Ouyang, L., Wu, J., Jiang, X., Almeida, D., Wainwright, C. L., Mishkin, P., Zhang, C., Agarwal, S., Slama, K., Ray, A., Schulman, J., Hilton, J., Kelton, F., Miller, L., Simens, M., Askell, A., Welinder, P., Christiano, P., Leike, J., and Lowe, R. (2022). Training language models to follow instructions with human feedback.
>
> [4] Chan, S. C. Y., Santoro, A., Lampinen, A. K., Wang, J. X., Singh, A., Richemond, P. H., McClelland, J., and Hill, F. (2022). Data Distributional Properties Drive Emergent In-Context Learning in Transformers.
>
> [5] Radford, A., Wu, J., Child, R., Luan, D., Amodei, D., and Sutskever, I. (2019). Language Models are Unsupervised Multitask Learners.
>
> > This fails to prove any practical comparative advantage.
>
> Improving on compositional generalization is an important step towards better AI, and we clearly show that our proposal does that.  In addition to our standard Transformer baselines (which our architecture surpass in all experiments) and comparisons with GPT-5 Thinking and GPT-4.1, we directly compare against a contemporary test-time adaptive, latent variable model specifically designed for ARC-AGI, namely Latent Program Network (Macfarlane and Bonnet, 2025). Our model performs significantly better (25.1% accuracy) vs LPN (1.9% accuracy) on our 1-D ARC task and 46.1% (ours) vs 37.1% (LPN) on SRAVEN.
>
> [the response continues in the next comment]

---

> ### Author Response · Authors · 2025-11-15
>
> >Use of Arbitrary Benchmarks: The authors used a niche, self-created benchmark called ""1-D ARC-LIKE REASONING"", which they only state is ""inspired by"" the standard 2D ARC benchmark.
>
> These benchmarks are not arbitrary; they are chosen to specifically evaluate compositional generalization.  For this reason, we choose to generate our train-test splits in 1-D ARC using composable combinator functions from the open source synthetic data generation tool [arc-like](https://github.com/neurallambda/arc-like). While the ARC-AGI benchmark offers a compelling test of abstract reasoning, it lacks explicit controllability for verifying the presence of compositional generalization in models. That is, the test problems do not consist of verifiable compositions of training problems, and it is thus difficult to isolate the effect of composition on solvability. This is also why we choose to evaluate on SRAVEN (Schug et al. 2025); it gives us the ability to control the train-test splits so that the test split contains problems that are precise subsets of possible compositions of training problems.
>
> >On the contrary, on the 2D ARC benchmark—which should be more difficult—some existing LLMs have achieved nearly 70% accuracy. How does this discrepancy arise?
>
> This discrepancy likely arises because the ARC-AGI training and evaluation sets, as well as many augmented versions of their training sets, are publicly available and therefore have been seen by LLMs during pretraining. LLMs are good at solving tasks for which they have already seen similar examples, but this ability does not involve compositional generalization.
>
> >Moreover, seeing that modern LLMs like GROK achieve nearly 0% on that benchmark suggests the benchmark has high variance between LLMs and is a task that LLMs are inherently bad at. Why does high performance here validate this model?
>
> We agree with the reviewer on this point;  there is significant room for LLMs to improve their compositional generalization ability. That is why we believe our contribution is of importance. We mention LLMs since many of the current SOTA methods in solving ARC-AGI problems are based on LLMs ([ARC Leaderboard](https://arcprize.org/leaderboard)). Our experiments identify compositional generalization as a weak-point in these SOTA models, further justifying an evaluation which focuses on precisely this issue.
>
> >For example, VLMs perform poorly at depth estimation. A very simple depth estimation model will easily outperform a VLM. What value does that simple model have?
>
> We believe compositional generalization to be a vital aspect of intelligence and that eliciting it in modestly sized transformers to this extent is a significant contribution.  Furthermore, we are not simply hand-crafting a model specifically to do composition; ours is a general-purpose architecture which we show has the properties needed to perform well when compositional generalization happens to be needed.
>
>
> >This merely proves the VLM's inability on that task; it does not guarantee the new model's performance. This appears to be an attempt to avoid direct comparison on recognized benchmarks, severely damaging the credibility and generalizability of the experimental results.
>
> We agree with the reviewer that a baseline’s inability to solve a task does not guarantee a proposed model’s ability. Therefore we do not only report the performance of the baseline, but also the performance of our model, which significantly outperforms the baseline architectures (see above response).
>
> >Unrealistic Inference Cost (Test-Time Training): The most significant problem is that the method performs 100 steps of gradient search (fine-tuning) to solve a single test query. This is not fast few-shot inference. This is, in effect, Test-Time Training. This approach of training a model for 100 steps on a single problem renders any comparison against MAML (which takes only a few steps) or GPT (which requires no extra training) fundamentally meaningless. This is an unrealistic inference cost and not a fair comparison."
>
> Because our model weights are never updated during test-time gradient search, and because the number of parameters in our latent representations we search over are orders of magnitude fewer than updating the whole model (10^3 vs 10^6), our approach cannot be compared to retraining a model. The wall-clock time for performing 100 steps of gradient search over a batch of 512 1-D ARC problems is ~14 seconds on a single H100 GPU, and is therefore a realistic inference cost. In comparison, in our experiments, GPT-5-Thinking expends up to 10,000 tokens (requiring 10,000 forward passes) during inference on a single problem,, using >100B parameters over context lengths 10^3 times longer than what our model takes as input.
>
> Thank you once again for taking your time to review our paper, it is very much appreciated.

---

> ### Comment · Reviewer_tPRa · 2025-11-25
> **Use of Arbitrary Benchmarks Criticisms contains Factual Errors**
>
> I would like to note that I completely disagree with the following statement by the reviewer VaZe, as it contains multiple factual errors. I believe all of these criticisms should be discounted when assessing this paper.
>
> "Use of Arbitrary Benchmarks: The authors used a niche, self-created benchmark called '1-D ARC-LIKE REASONING,' which they only state is 'inspired by' the standard 2D ARC benchmark. A search for this benchmark reveals it is extremely niche. Furthermore, even the standard 2D ARC benchmark is not frequently used. On the contrary, on the 2D ARC benchmark—which should be more difficult—some existing LLMs have achieved nearly 70% accuracy."
>
> First, the statement that 2D ARC is not frequently used is completely false. On the contrary, ARC-AGI is one of the most popular benchmarks in current AI research that has proved to be a huge weakness for LLMs and which has been a north star for guiding reasoning LLM iterations. The ARC-AGI 2024 technical report has over 100 citations and performance on the benchmark is regularly released in the model cards for frontier LLMs.
>
> Second, the statement that "LLMs have achieved nearly 70% accuracy" completely lacks context regarding the inference strategy used and total compute. This statement cannot just be made in isolation, and it is only useful to compare when inference compute is the same.
>
> I also note that the review makes an incorrect citation of 0% Grok accuracy. In fact, Grok 4 (released July 9th, 2025) is one of the highest performing LLMs (66.7% with thinking on ARC-AGI 1).

---

> > ### Comment · Reviewer_VaZe · 2025-11-26
> > **Clarification,**
> >
> > The word 'niche' is  simply about 1-D AGI benchmark.

---

> > > ### Comment · Reviewer_VaZe · 2025-11-26
> > > **Remark**
> > >
> > > Comparison to LLM is meaningless. For example, nobody says calculator has better reasoning capability sinc it beats GPT-5 at arithmetic.  If specialized model beats 1D arc Pattern, It cannot say it has better generalization ability.

---

> ### Comment · Reviewer_tPRa · 2025-11-25
> **Unrealistic Inference Cost (Test-Time Training) Comments**
>
> The reviewer notes:
>
> "Unrealistic Inference Cost (Test-Time Training): The most significant problem is that the method performs 100 steps of gradient search (fine-tuning) to solve a single test query. This is not fast few-shot inference. This is, in effect, Test-Time Training. This approach of training a model for 100 steps on a single problem renders any comparison against MAML (which takes only a few steps) or GPT (which requires no extra training) fundamentally meaningless. This is an unrealistic inference cost and not a fair comparison."
>
> This criticism does not have much basis in my view. Each method has different inference costs. LLMs, when not using Chain-of-Thought, do not leverage increased compute to increase performance. A comparison is still valid, as this is a strategy of inference. In addition, there is nothing the authors can do about the fact that MAML only leverages a few gradient steps at test time. The key is that MAML precisely cannot leverage additional steps without overfitting. The authors propose a method that can leverage additional compute with additional performance. While equalising compute where possible is preferable, I believe the reviewers have done a reasonable job. Note that comparisons to LLMs are arbitrary anyway due to vastly different model sizes and training cost assumptions, and so equalising compute there is a meaningless pursuit.

---

> ### Comment · Reviewer_VaZe · 2025-11-26
> **Requirement Clarification.**
>
> Your claim: "Compositional generalization through nonparametric latent space"
>
> Required experiments to support this claim:
> - TieredImageNet with frozen ResNet backbone
> - MiniImageNet with frozen ResNet backbone
>
> These are STANDARD benchmarks in the few-shot learning literature. Your architecture is fully capable of handling
> these tasks - you use encoder-decoder transformers, which can process images
> and language just fine.
>
> Your current experimental suite:
> - 1-D ARC (self-created, ~10 GitHub stars)
> - SRAVEN (compositional, but symbolic only)

---

> > ### Comment · Reviewer_VaZe · 2025-11-26
> > **the algorithm can be extended to 2d-AGI**
> >
> > Your architecture claims to be general-purpose
> >
> > Setting:
> > encoder: Standard transformer encoder (tokenizes any input)
> > decoder: Standard transformer decoder (generates any output)
> > method: Works on sequences
> >
> > 2D ARC implementation requirements:
> > 1. Tokenize 2D grids → flatten to sequence or use 2D positional encoding
> > 2. Encoder processes tokens → IDENTICAL to your current setup
> > 3. Decoder generates output → IDENTICAL to your current setup
> >
> > This is a TRIVIAL extension. Papers routinely handle both 1D and 2D:
> > - Vision Transformers: Handle 2D images via patch tokenization
> > - Your own SRAVEN experiments: Handle 2D matrix structures
> > - Standard practice: Test on multiple input modalities
> >
> > You implement SRAVEN (2D matrix reasoning) but not 2D ARC?
> >
> > 1D-AGI is non-trivial benchmark within <10 github stars.

---

### Official Review · Reviewer_tPRa · 2025-10-31

**Soundness:** 3
**Presentation:** 3
**Contribution:** 3
**Rating:** 6
**Confidence:** 4

**Summary:**

The authors build on recent work (LPN) that investigates learning compressed latent spaces that can be searched at test time to reduce the amortization gap from amortized inference. In this work, the paper proposes the Abduction Transformer, which uses multiple mixture distributions (via Dirichlet Processes), instead of a single Gaussian distribution, to learn the approximate posterior distribution. The method incorporates information-theoretic regularization and test-time gradient-based search in a nonparametric latent space. It is evaluated on OOD compositional meta-learning tasks such as ARC-like program induction, Raven’s progressive matrices, and linguistic systematicity tasks.

**Strengths:**

- Targets a critical weakness in LPN: which does not necessarily handle compositionality in the latent space due to a single latent vector.
- Approach to tackle compositionality is simple and scalable
- Explores performance on a range of experiments, including ARC-like program induction, Raven’s progressive matrices, and linguistic systematicity tasks.
- Uses appropriate baselines, like single-vector LPN, standard transformers, and GPT-5 Thinking.

**Weaknesses:**

- Lack of ablations on the method. Since the Abduction Transformer performance is supposedly derived from the larger number of distributions compared to LPN it is important to perform an ablation that scales this axis to see when compositionality emerges. Specifically, a graph with number of distributions on x-axis and test-time gradient search and non-gradient search performance plotted in the graph.

- There is no evaluation on non-compositional data.

**Questions:**

1. Is it possible to control the number of distributions in the method? If so, how does performance of your method scale with number of distributions starting from 1 and increasing beyond what you investigated, to greater understand what is the key component generating compositionality. Any additional / alternative ablations that could give more insight into this would strengthen the paper also.

2. How does performance compare on non-compositional datasets or datasets that do not explicitly target the requirement of compositionality?

3. Can you give examples of problems solved and failed by LPN and the Abduction Transformer to understand more clearly the failure and success modes of each?

---

> ### Author Response · Authors · 2025-11-15
>
> We thank the reviewer for their feedback and positive comments.
>
> > Lack of ablations on the method. Since the Abduction Transformer performance is supposedly derived from the larger number of distributions compared to LPN it is important to perform an ablation that scales this axis to see when compositionality emerges. Specifically, a graph with number of distributions on x-axis and test-time gradient search and non-gradient search performance plotted in the graph.
>
> We ablate the set-of-vector approach used by Abduction Transformer by directly comparing it to the standard LPN, which uses a single vector. Hence, this serves as a many vs one ablation on the number of components in the representation. A key benefit of using a DP as our distribution over latent representations is that in cases where fewer components are needed, the DP can assign small probability over components which are not needed. Our implementation of denoising attention discretely drops such low probability components, dynamically modifying the number of integer components depending on problem complexity.
>
> > There is no evaluation on non-compositional data.
>
> In this work, we highlight the fact that while current Transformer based architectures are capable of a wide range of tasks, compositional generalization is a critical weak point where such models show particularly severe degradation in performance. For that reason, we focus on the domain of compositional problems, as improved performance in this domain is our main contribution and what differentiates Abduction Transformer from other methods.
>
> > Is it possible to control the number of distributions in the method? If so, how does performance of your method scale with number of distributions starting from 1 and increasing beyond what you investigated, to greater understand what is the key component generating compositionality. Any additional / alternative ablations that could give more insight into this would strengthen the paper also.
>
> Thank you for this suggestion. Currently the model learns to choose how many vectors it needs to represent a given input. However, we could add a constraint to force it to drop more vectors, and see how this affects compositional generalization. Our concern is that the resulting degradation might reflect more the way we implement the constraint than any fundamental property of the model, but this should become evident by seeing if it converges to the LPN baseline. We will try this and include it in the paper if it provides meaningful insights.
>
> > How does performance compare on non-compositional datasets or datasets that do not explicitly target the requirement of compositionality?
>
> We did not evaluate our model on non-compositional datasets for reasons explained in our response to your second comment, “There is no evaluation on non-compositional data.“
>
> [the response continues in the next comment]

---

> > ### Comment · Reviewer_tPRa · 2025-11-21
> >
> > I appreciate the authors’ focus on compositional generalisation, but I maintain my belief that evaluation on non-compositional data remains essential to interpret the results correctly.Without such a comparison, we cannot rule out the possibility that the Abduction Transformer simply outperforms LPN on any task, for reasons unrelated to compositionality (e.g., better optimization, higher capacity, or inductive biases).

---

> ### Author Response · Authors · 2025-11-15
>
> > Can you give examples of problems solved and failed by LPN and the Abduction Transformer to understand more clearly the failure and success modes of each?
>
> For 1-D ARC (containing 2048 problems), out of the 14 unseen composition types contained in the test set, Abduction Transformer solves at least one problem for 10/14 of them, while LPN solves at least one problem for 4/14. Abduction Transformer is able to solve all compositions that LPN is able to solve. Abduction Transformer is able to solve problems containing hypotheses that are compositions of 3 transformations, while LPN is not. Error analysis is over 1 seed. All listed transformations are shown visually in Fig. 5 in App. D. We are happy to discuss this error analysis in the paper.
>
> **Abduction Transformer (ours)**
>
> | Puzzle                                        |   Perfect Solves |   Total |   Perfect Rate |
> |:----------------------------------------------|-----------------:|--------:|---------------:|
> | colorshift(2)_endpoints                       |              135 |     148 |         0.9122 |
> | magnets(2)_colorshift(1)                      |              109 |     143 |         0.7622 |
> | extend_to_pivot_colorshift(1)                 |               71 |     134 |         0.5299 |
> | translate(1)_denoise                          |               68 |     159 |         0.4277 |
> | translate(2)_colorshift(1)_expand(1)          |               28 |     150 |         0.1867 |
> | translate(1)_remove_shortest_blocks_endpoints |               11 |     141 |         0.0780 |
> | move_to_pivot_translate(2)                    |               11 |     168 |         0.0655 |
> | translate(4)_translate(2)                     |                6 |     139 |         0.0432 |
> | expand(1)_repaint_max_block_translate(2)      |                5 |     144 |         0.0347 |
> | expand(2)_reflect(seq_len//2)                 |                2 |     157 |         0.0127 |
> | reflect_pivot_expand(1)                       |                0 |     142 |         0.0000 |
> | rotate_block_pixels(1)_reflect(seq_len//2)    |                0 |     142 |         0.0000 |
> | translate(2)_reflect_pivot_translate(1)       |                0 |     142 |         0.0000 |
> | endpoints_expand(2)_shrink                    |                0 |     139 |         0.0000 |
>
> **LPN**
> | Puzzle                                        |   Perfect Solves |   Total |   Perfect Rate |
> |:----------------------------------------------|-----------------:|--------:|---------------:|
> | colorshift(2)_endpoints                       |                7 |     148 |         0.0473 |
> | magnets(2)_colorshift(1)                      |                0 |     143 |         0.0000 |
> | extend_to_pivot_colorshift(1)                 |                1 |     134 |         0.0075 |
> | translate(1)_denoise                          |               17 |     159 |         0.1069 |
> | translate(2)_colorshift(1)_expand(1)          |                0 |     150 |         0.0000 |
> | translate(1)_remove_shortest_blocks_endpoints |                0 |     141 |         0.0000 |
> | move_to_pivot_translate(2)                    |                2 |     168 |         0.0119 |
> | translate(4)_translate(2)                     |                0 |     139 |         0.0000 |
> | expand(1)_repaint_max_block_translate(2)      |                0 |     144 |         0.0000 |
> | expand(2)_reflect(seq_len//2)                 |                0 |     157 |         0.0000 |
> | reflect_pivot_expand(1)                       |                0 |     142 |         0.0000 |
> | rotate_block_pixels(1)_reflect(seq_len//2)    |                0 |     142 |         0.0000 |
> | translate(2)_reflect_pivot_translate(1)       |                0 |     142 |         0.0000 |
> | endpoints_expand(2)_shrink                    |                0 |     139 |         0.0000 |
>
> Thank you once again for taking your time to write this review, it is very much appreciated.

---

> ### Author Response · Authors · 2025-11-28
> **Experiments on Non-compositional Data**
>
> Thanks for the useful feedback!
>
> We appreciate your suggestion regarding including non-compositional evaluations to demonstrate that our method's advantage indeed comes from better compositional generalization. To this end, we present evaluation results on 1-D ARC and SRAVEN for both LPN and Abduction Transformer where the test set is in-distribution relative to the training set, i.e, it does not contain any unseen compositions. This evaluation thus measures the ability of LPN and Abduction Transformer to solve 1-D ARC and SRAVEN problems which do NOT require compositional generalization.
>
> We evaluate 2048 test problems on the same model checkpoints used for our evaluations in §2 and §3 and report accuracy over gradient step counts.
>
> **1-D ARC (Non-compositional)**
>
> LPN (Baseline)
> | Gradient Steps | 0      | 10     | 100    |
> | -------------- | ------ | ------ | ------ |
> | Solve Rate     | 97.66% | 98.05% | 97.90% |
>
>
> Abduction Transformer (Ours)
> | Gradient Steps | 0      | 10     | 100    |
> | -------------- | ------ | ------ | ------ |
> | Solve Rate     | 97.36% | 98.39% | 98.39% |
>
> **SRAVEN (Non-compositional)**
>
> LPN (Baseline)
> | Gradient Steps | 0      | 10     | 100    |
> | -------------- | ------ | ------ | ------ |
> | Solve Rate     | 99.85% | 99.90% | 99.90% |
>
>
> Abduction Transformer (Ours)
> | Gradient Steps | 0      | 10     | 100    |
> | -------------- | ------ | ------ | ------ |
> | Solve Rate     | 99.95% | 99.95% | 99.95% |
>
> These results show that when compositional generalization is NOT required, LPN closely matches Abduction Transformer's ability to solve new problems at test time. We believe this shows that the superior performance of our method on OOD composition tasks (in  §2 and §3)  comes specifically from better compositional generalization ability.

---

### Official Review · Reviewer_1Rbo · 2025-10-31

**Soundness:** 2
**Presentation:** 4
**Contribution:** 1
**Rating:** 2
**Confidence:** 4

**Summary:**

This work addresses program synthesis tasks, with a focus on out-of-distribution (OOD) and compositional generalization. They introduce a new architecture named the Abduction Transformer, whose goal is to infer a latent cause (hence, abduction) of the observed specification for a given task, and use it to predict the solution of the considered task. This two-stage process implicitly defines a probabilistic graphical model corresponding to a deep variational Bayesian model, of the kind used in variational autoencoders (VAEs).

The main contributions of the paper include:
- a VAE formulation for neural program synthesis
- the representation of the latent space as a nonparametric Dirichlet Process (DP)
- gradient-based search in the latent space to maximize the likelihood of the specification
- training with 1 gradient step, using the ELBO loss

The authors experiment with the Adbuction Transformer on three different benchmarks: a 1D ARC dataset, Raven's Progressive Matrices, and a linguistic systematicity task. They highlight performance improvements due to latent search and some OOD generalization at test-time.

**Strengths:**

The paper is well written, the method is promising, and the experimental setups are somewhat well explained. I particularly enjoyed Figure 1, which details the method very well (although the DP equations could be skipped in this figure for higher clarity). In my opinion, the largest novelty in this work is the modeling of the latent space as a DP.

I believe the paper tackles a crucial aspect of machine learning, i.e., OOD generalization in the context of program synthesis, and demonstrates the proposed architecture quite well with a set of 3 different benchmarks. The related work section successfully connects the Abduction Transformer to recent works such as Test-Time Fine-Tuning [Hübotter et al., 2025] or the Latent Program Network [Macfarlane & Bonnet, 2025]. Overall, the work tackles an important problem in machine learning, and the proposed architecture seems promising to overcome some of the limitations of current SOTA systems.

**Weaknesses:**

There are several weak points I would like to address in this review.

## Novelty
The authors carefully relate this work to the framework of VAEs [Kingma et al., 2013] and the baseline Latent Program Network (LPN) [Macfarlane & Bonnet, 2025]. After carefully analyzing the [LPN paper](https://arxiv.org/abs/2411.08706) and comparing it to this work, it is not clear to me what novelty this work brings that is not covered in the LPN paper. This is the most important weakness to me; it seems that the LPN paper covers most of what is "introduced" in this work, in its current version.
- VAE network for program synthesis: in LPN.
- Training loss: identical to LPN, i.e., max likelihood and KL to prior.
- Training dynamic: 1-step of gradient descent in latent space, no backprop through latent update.
- Test-time search: gradient descent in latent space, maximizing likelihood of the specification.

A notable difference from the LPN baseline is the DP process used to parameterize the latent space (the baseline is a standard normal Gaussian distribution). If this is an important novel contribution from this paper, I would like the authors to demonstrate it (see ablations below).

## Experiments
The three experiment setups presented in this paper remain on the toy level. Since ARC-AGI is mentioned, it would improve the contributions of this paper to demonstrate the generalization abilities of the Abduction Transformer on the ARC-AGI v1 or v2 benchmark. Alternatively, the [ConceptARC benchmark](https://arxiv.org/abs/2305.07141) could be a more accessible version to try it on.

## Baselines
In addition to the LPN baseline, it would improve the paper to include other program synthesis baselines, especially if ConceptARC or ARC-AGI is considered to be added to the experiments.

## Ablations
The parameterization of the latent space as a DP seems interesting and would need to be ablated to be justified. The number of latent space gradient steps during training does not seem to be ablated or justified either.

**Questions:**

1. What made you go for a DP to model the latent space? What justifies moving from the standard implementation of Kingma et al., 2013?

2. What are the novel aspects introduced in this work that you would like to emphasize?

---

> ### Author Response · Authors · 2025-11-15
>
> Thank you for the detailed and effortful review, we appreciate your time investment. We appreciate that you find our paper to be well written. We will address your feedback below.
>
> > it is not clear to me what novelty this work brings that is not covered in the LPN paper. This is the most important weakness to me; it seems that the LPN paper covers most of what is "introduced" in this work, in its current version.
>
> Thank you for this question; it is indeed a very important point which we have made clearer in an updated version of the paper, and answer here.  While it is true that LPN introduces the idea of using a VAE for program synthesis with latent space search, our approach fundamentally differs in that LPN has a fixed-dimensional vector latent space, whereas our model’s latent space is a set of weighted vectors whose size can grow with the required complexity.  This fundamental difference leads to a number of novelties:
> - LPN infers a distribution over vectors as a Gaussian, while we infer a distribution over mixture distributions (i.e. sets of weighted vectors) as a Dirichlet Process
> - This enables our approach to represent compositional problems as sets of vectors whose size can vary according to the compositional complexity of the input, whereas LPN represents problems as a single, fixed size vector that cannot adapt to increased complexity
> - The original LPN paper focuses on generic capabilities on program synthesis tasks, while we focus on explicitly measuring compositional generalization across abstract spatial, symbolic and linguistic tasks by testing on datasets containing verifiable compositions of those seen during training.
> - Our approach performs significantly better on 1-D ARC for OOD compositions: 25.1% accuracy (ours) vs 1.9% accuracy (LPN), and 46.1% (ours) vs 37.1% (LPN) on SRAVEN.
>
> > Since ARC-AGI is mentioned, it would improve the contributions of this paper to demonstrate the generalization abilities of the Abduction Transformer on the ARC-AGI v1 or v2 benchmark.
>
> Thank you for this suggestion. Our aim in this paper was to explicitly demonstrate an architecture capable of compositional generalization, which requires us to be able to verify that the problems in the test set are compositions of those seen during training. This is not possible for ARC-AGI v1 and v2, as there is no rigorous compositional relationship between problems in the train-test splits. To this end, we chose to focus on a dataset where compositionality can be precisely controlled. We have made this point clearer in an updated version of the paper.
>
> > In addition to the LPN baseline, it would improve the paper to include other program synthesis baselines, especially if ConceptARC or ARC-AGI is considered to be added to the experiments.
>
> Since our focus is on compositional generalization rather than program synthesis in particular, we did not include architectures which are tailored specifically for program synthesis (such as those using search over domain specific languages) that lack the universality that our method, LPN, and other transformer baselines have.
>
> > The parameterization of the latent space as a DP seems interesting and would need to be ablated to be justified.
>
> Thank you for raising this point; we are sorry if this was not clear in the original submission. The LPN baseline is in fact an ablation of our model where the DP has been replaced with a Gaussian.  We reimplemented the LPN model to ensure that our model only differs in this way.
> As you point out, our model differs from the LPN model only in that the LPN’s Gaussian distribution over latent vectors is replaced with our model’s DP distribution over mixture distributions, and our experiments show significantly improved performance when using a DP (see response to your first comment). Furthermore, we ablate the KL-regularization in the DP and show that performance decreases without it. We also show that naively searching over sets of vectors without treating them as samples from a DP leads to much poorer performance (encoder-decoder transformer baseline in Table 1 and 2), indicating that the specific formulation of latent representations as samples from DPs is necessary to achieve our improved results.
>
> [the response continues in the next comment]

---

> > ### Author Response · Authors · 2025-11-15
> >
> > > The number of latent space gradient steps during training does not seem to be ablated or justified either.
> >
> > Thank you for raising this point. We found that increasing the number of gradient steps during training of our model led to faster convergence but not an improvement in absolute performance on evaluation sets. Furthermore, (Macfarlane and Bonnet, 2025) show that LPN has the best performance on ARC tasks when trained with 1 gradient step during training, as opposed to 0 steps or training with 2 or more steps. We added this point in our updated version of the main text.
> >
> > > What made you go for a DP to model the latent space? What justifies moving from the standard implementation of Kingma et al., 2013? What are the novel aspects introduced in this work that you would like to emphasize?
> >
> > Please see our response to your first comment. We would like to emphasize that extending VAEs’ latent space from fixed-dimensional vectors to attention-based set-of-vector representations has been a major advance over the past couple years, and our paper is further progress in this trend.
> >
> > We thank the reviewer for their time and feedback.

---

### Official Review · Reviewer_xg9b · 2025-10-31

**Soundness:** 2
**Presentation:** 3
**Contribution:** 2
**Rating:** 4
**Confidence:** 3

**Summary:**

The authors propose the Abduction Transformer to achieve better compositional generalisation on out-of-distribution (OOD) meta-learning tasks. The key idea is to represent each few-shot task with a nonparametric latent hypothesis, modelled as a Dirichlet-process (DP) mixture over latent vectors produced by the encoder. This allows the model to adaptively allocate a variable number of latent components to capture the compositional structure of the task. The authors evaluate their approach on three tasks: 1-D ARC-like program induction, symbolic Raven's Progressive Matrices (SRAVEN), and linguistic systematicity and show competitive performance with fewer parameters (1.2M), outperforming standard transformer baselines and comparable to GPT-5 Thinking on certain tasks.

**Strengths:**

- The paper has a clear probabilistic framing, casting few-shot meta-reasoning as posterior inference over a task-specific hypothesis, trained with a variational free-energy bound and refined by test-time gradient search.
- The DP prior over the latent space acts as a strong architectural prior that naturally aligns with the compositional learning premise.
- The authors show the results of extensive ablation studies on modeling choices such as KL-regularisation over the DP posterior and test-time gradient search, indicating both are crucial.
- The latent-space visualization suggests unseen compositions embed near their constituent primitives, which is qualitatively consistent with the paper’s goal of compositional reuse.

**Weaknesses:**

- From the paper, it seems that there are three types of composition: in data (function composition) as in 1-D ARC, in hypothesis space (production) as with the linguistic systematicity, and in features as independent submodules (parallel composition) as in SRAVEN. The issue is that the paper conflates these fundamentally different kinds of composition under a single formalism. Although all are expressed with the same symbolic notation $H_1(H_2(x))$, they correspond to distinct cognitive and computational challenges. Without explicitly distinguishing these regimes, it becomes unclear whether the reported improvements arise from genuine compositional reasoning or simply from learning parallel or modular mappings.
- The paper relies on approximate KL and truncating DP samples ($\kappa_0 = n + 1$), but does not quantify the approximation or truncation error in training or test-time search. Without rates of approximation or stability results, the “information-theoretic regularisation” claim remains empirical.
- Why does gradient descent on latent representations lead to better hypotheses? It lacks convergence analysis and could benefit from more insights into the geometry and identifiability of the latent optimization: when and why gradient updates align with true compositional structure rather than overfitting to spurious patterns.
- The method could benefit from even a small-scale analysis on a real compositional regime such as code editing, or some visual entities from ARC-AGI. Would it be possible to show such results?
- The baseline comparisons for methods that perform test-time adaptation are limited to just one: single vector LPN and there are no comparisons to other compositional generalisation methods (for some examples which might not be up to date, neural module networks, program synthesis approaches, neural interpreters etc). Even if some methods need adaptation or don't directly apply, discussing why would strengthen the paper.
- The notation is confusing since the paper uses $H$ both as a _function_ (the hypothesis mapping in compositional generalisation) and as a _latent variable_ (a mixture distribution in the nonparametric encoder). This can be made clearer by using, for eg, $h$ as the latent variable and $H_h(\cdot)$ as the functional mapping.
- The writing of the central claim can be made a bit clearer in my opinion. The 1-D ARC and SRAVEN setups are synthetic, controlled, and compelling for diagnosis, but the core gains are still modest, and comparisons to “GPT-5 Thinking” are zero-shot and prompt-based versus a trained model with test-time search making not like-for-like in compute or adaptation.
- Appendix B claims denoising attention "subsumes" regular attention for _discrete distributions_ (categorical variables with additive noise), not for the _continuous Gaussian_ case used in the model. Without specifying assumptions like zero noise variance or small-variance limits, it is not clear the conditions under which this holds.


I am happy to update the score of the paper if my questions and concerns can be addressed.

**Questions:**

- Under what assumptions on the decoder does the test-time objective exhibit a landscape such that gradient search converges to semantically correct hypotheses rather than spurious modes?
- It's interesting to see the model maintain high accuracy even on less than 7 samples from the linguistics task. How can you disentangle prior-based guesses from genuine compositional recovery?
- Maybe I am missing something but can you explain why the number of gradient steps while training is 1 but is 100 for evals?
- Why is it appropriate to average samples from different distributions at the encoder's output? Why is averaging done here rather than using a product of experts or other combination method?

---

> ### Author Response · Authors · 2025-11-15
>
> Thank you for writing this detailed review, we appreciate your feedback.
>
> >it seems that there are three types of composition: in data (function composition) as in 1-D ARC, in hypothesis space (production) as with the linguistic systematicity, and in features as independent submodules (parallel composition) as in SRAVEN. The issue is that the paper conflates these fundamentally different kinds of composition under a single formalism.
>
> Thank you for raising this point, we do in fact explicitly disentangle and give precise definitions for function composition and production composition in §2. Furthermore, each experimental setup explicitly states which notion of compositional generalization we are testing for.
>
> >it becomes unclear whether the reported improvements arise from genuine compositional reasoning or simply from learning parallel or modular mappings.
>
> We agree that fundamental objective is genuine compositional reasoning, and this is why we choose to define compositional generalization simply as the ability to solve problems which are unseen compositions (according to the definitions of composition in §2) of those seen during training.  We intentionally subsume learning both parallel and modular mappings in this ability, because we want to evaluate the general ability to do compositional reasoning.  Our experiments show that our method achieves compositional generalization better than LPN (Macfarlane and Bonnet, 2025) and standard transformer architectures, without being tailored to any specific form of composition.
>
> > The paper relies on approximate KL and truncating DP samples, but does not quantify the approximation or truncation error in training or test-time search. Without rates of approximation or stability results, the “information-theoretic regularization” claim remains empirical.
>
> The soundness of the KL approximation and truncation procedure we use for DP samples has been addressed in previous work (Henderson and Fehr, 2023), and is not the topic of this paper. We leverage these previous theoretical results to provide a theoretical justification for our architectural choices, and contribute additional empirical results further validating these justifications.
>
> > Why does gradient descent on latent representations lead to better hypotheses? It lacks convergence analysis and could benefit from more insights into the geometry and identifiability of the latent optimization
>
> We agree that these properties of the induced latent hypothesis space are crucial to the success of our model.  While we do not provide any theoretical analysis of these properties, our empirical results do address them.  Firstly, the improvements gained with gradient-based search indicate that our induced latent space is smooth enough to be searchable.  This conclusion is reinforced by our experiments on scaling gradient steps in Fig. 6 (in Appendix), which show that increasing gradient steps leads to log linear improvements in performance, suggesting that they are converging to better hypotheses.  Also, we provide visualizations of our latent space after training with gradient search in Fig. 7 (in Appendix), which demonstrate that our training method with gradient search leads to a well separated geometric space over hypotheses. Furthermore, unseen compositions are mapped to locations in between their compositions, indicating that the learned geometry has compositional semantics. We take this as an important aspect of why our model achieves compositional generalization.
>
> > The method could benefit from even a small-scale analysis on a real compositional regime such as code editing, or some visual entities from ARC-AGI. Would it be possible to show such results?
>
> Thank you for raising this point. While ARC-AGI is an interesting test of general abstract reasoning, it does not allow us to measure compositional generalization ability, since the problems contained in the benchmark are not verifiable compositions of primitive operations. We therefore chose benchmarks which allow us to precisely control the train-test splits so that problems in the test set are guaranteed to be unseen compositions of problems seen during training.
>
> [the response continues in the next comment]

---

> > ### Author Response · Authors · 2025-11-15
> >
> > > The notation is confusing since the paper uses both H as a function (the hypothesis mapping in compositional generalisation) and as a latent variable (a mixture distribution in the nonparametric encoder). This can be made clearer by using, for eg, h as the latent variable and H_h as the functional mapping.
> >
> > Thank you for this feedback. This is a good point and we will modify our notation to make the distinction between the latent variable representing the hypothesis and the function it executes clearer throughout our paper.
> >
> > > The writing of the central claim can be made a bit clearer in my opinion. The 1-D ARC and SRAVEN setups are synthetic, controlled, and compelling for diagnosis, but the core gains are still modest, and comparisons to “GPT-5 Thinking” are zero-shot and prompt-based versus a trained model with test-time search making not like-for-like in compute or adaptation.
> >
> > The comment regarding comparisons with GPT is a good point, and we appreciate it being raised. We already make clear that the comparisons to GPT-5 are nominal and without fine-tuning, but we have made this clearer in the updated version of the main text.
> >
> > However, we would like to emphasize that the gains in performance of our method over LPN (Macfarlane and Bonnet, 2025) and transformer baselines on 1-D ARC are 25.1% (ours) vs < 6% (LPN and other baselines), and this goes beyond what can be considered ‘modest gains’.
> >
> >
> > >Appendix B claims denoising attention "subsumes" regular attention for discrete distributions (categorical variables with additive noise), not for the continuous Gaussian case used in the model. Without specifying assumptions like zero noise variance or small-variance limits, it is not clear the conditions under which this holds.
> >
> > Denoising attention is a generalization of regular attention in the sense that any regular attention function over any set-of-vectors can be mapped to an equivalent denoising attention function over discrete mixtures of impulse distributions.  In our model, our latent representations are also discrete mixture distributions; they are only sampled from a DP parameterized by continuous Gaussian mixtures. Hence, the denoising attention operation applied to our latent representations is in effect equivalent to regular attention, but with biases added to the attention weights to encode the mixture weights (which is shown by Henderson and Fehr, 2023). This holds regardless of assumptions on variance in the DP’s Gaussian mixture.
> >
> > > Under what assumptions on the decoder does the test-time objective exhibit a landscape such that gradient search converges to semantically correct hypotheses rather than spurious modes?
> >
> > The KL-compression objective during training encourages the hypothesis space to be well separated (as shown in Fig. 7), and contain simple, low-complexity representations. If a representation in this space achieves low loss on the few-shot examples, it is likely to be semantically correct, since spurious modes which depend on hypotheses with high complexity (overfitting) are unlikely to be found in this space.  The noise introduced by NVIB trains the decoder to be robust to this noise by providing a smooth interpretation to this space, thereby making it searchable with the gradient of the decoder.  We don’t believe there are any specific properties required for the decoder, other than being differentiable and powerful enough to compute the desired mapping.  But we would be very interested if the reviewer has suggestions for how to better characterize the requirements for the decoder.
> >
> > > It's interesting to see the model maintain high accuracy even on less than 7 samples from the linguistics task. How can you disentangle prior-based guesses from genuine compositional recovery?
> >
> > This is an interesting question. We believe that below 7 samples, the contributions on the DP posterior from the observed examples become relatively weak compared to the prior DP component, which represents prior beliefs about interpretation grammars before making any observations. Hence, our model is not only able to seamlessly adapt across varying degrees of complexity in the input, but also adapt to varying degrees of omission of information, by flexibly placing more weight on the prior representation when few-shot examples are scarce. I,e, when there is sufficient information we get ‘genuine compositional recovery’; when there is not enough information our model smoothly transitions to prior-based inference.
> >
> > [the response continues in the next comment]

---

> > > ### Author Response · Authors · 2025-11-15
> > >
> > > > Maybe I am missing something but can you explain why the number of gradient steps while training is 1 but is 100 for evals?
> > >
> > > We simulate 1 step of gradient search during training so that the encoder learns to infer initial hypotheses that are good initializations for the search process. We could in principle train with 100 steps of gradient search, but we found that training with more than one step of gradient search did not improve test-time performance. Macfarlane and Bonnet, 2025 also show that for LPNs, training with 1 step is better than training with 0 or > 1 steps. We have improved the discussion of this point.
> > >
> > > > Why is it appropriate to average samples from different distributions at the encoder's output? Why is averaging done here rather than using a product of experts or other combination method?
> > >
> > > Conceptually, it is appropriate to average the inferred latent representations because they each represent a particular estimate of the true hypothesis given a particular input/ouput pair. In other words, they are all estimates of the same thing, so averaging over them acts as an ensemble over estimators. We choose averaging over other methods because it is the most simple, theoretically sound way to aggregate hypotheses. It is important that the merging be done in the latent hypothesis space, since we then use the result to initialize search in this latent hypothesis space.
> > >
> > > Thank you very much for your detailed review.

---

### Official Review · Reviewer_3Yws · 2025-11-04

**Soundness:** 2
**Presentation:** 3
**Contribution:** 3
**Rating:** 8
**Confidence:** 2

**Summary:**

The paper introduces the Abduction Transformer (AT), a variational, test‑time‑adaptive architecture for systematic compositional generalization. AT encodes each few‑shot episode into a nonparametric latent space—a Dirichlet‑process (DP) mixture over a set of vectors—regularized by an information‑theoretic KL to a DP prior. A transformer decoder then cross‑attends to this latent mixture via denoising attention to generate outputs. Crucially, at test time the model refines the latent hypothesis $H$ by gradient descent on the few‑shot examples before answering the query, turning inference into iterative hypothesis testing. Across three tasks including 1‑D ARC‑like program induction, symbolic Raven’s (SRAVEN), and a linguistic systematicity benchmark, AT surpasses standard transformer baselines and a prior test‑time latent‑search model, and is competitive with zero‑shot GPT models under the authors’ protocol.

**Strengths:**

* This paper proposes a clear mechanism for compositionality through the procedure of (1) encoding few‑shot pairs, (2) inferring a DP posterior over latent hypotheses, (3) sampling a mixture $H$, (4) decoding, (5) refining $H$ with gradients on the few‑shot loss, and (6) answering the query. This operationalizes “search over hypotheses” rather than memorization.
* t‑SNE visualizations in fig. 7 show clear clustering of primitives and sensible placement of unseen compositions near their components, consistent with the intended semantics of the latent hypothesis space.
* Instead of a single latent vector, AT projects encoder outputs into DP parameters $(\mu,\sigma^2,\alpha)$ and samples a mixture with a variable effective number of components, which is an upgrade from fixed-size latents and aligns with the transformer’s token‑proportional outputs.
* Strong OOD results on diverse tasks are achieved with small models. AT achieves 25.1% perfect solve on strictly OOD 1‑D ARC compositions and 46.1% on SRAVEN with only 1% of rule combinations seen during training. The results match or surpass that from GPT-5 (Thinking) with only a 1.2M parameter budget.

**Weaknesses:**

* It is not very clear why denoising-attention is used in the AT decoder, and how much of the performance gain should be attributed to this design.
* The comparison to GPT-5 can be a bit unfair, as it uses a "low effort" setting, while a large reasoning token budget is quite important for getting good performance in such reasoning tasks. Also, being able to update the latent hypothesis at test time could be a huge advantage.

**Questions:**

* In table 2, why does the gradient-based refinement of $H$ harms the solve rate when trained on 90% of the possible compositions?
* In table 1 and 2, how does the performance of GPT scale with the reasoning token budget?
* How is the computational overhead compared to baseline methods such as Single Vector LPN?

---

> ### Author Response · Authors · 2025-11-15
>
> Thank you for your feedback and positive comments, we appreciate you recognizing the significance of our contributions.
>
> > It is not very clear why denoising-attention is used in the AT decoder, and how much of the performance gain should be attributed to this design.
>
> We use denoising attention because the latent representation that the decoder needs to access is a discrete mixture distribution. Denoising attention is equivalent to regular attention when the weights of that mixture are proportional to the L2-norms of the components (Henderson and Fehr, 2023). Since we would like our mixture weights to be able to take on arbitrary probabilities, we use denoising attention, which is a generalization of regular attention in which the restriction on mixture weights is removed.
>
> Our ablations ‘Abduction Transformer (No gradient search)’ for 1-D ARC and SRAVEN presented in Table 1, 2, both perform significantly worse than our method: 0.1% (ablation) vs 25.1% (ours) for 1-D ARC and 20.9% (ablation) vs 46.1% (ours). In this ablation, we do not apply our search procedure, but the model still uses denoising attention over latent representations. This shows that denoising attention on its own does not lead to compositional generalization.
>
> > The comparison to GPT-5 can be a bit unfair, as it uses a "low effort" setting, while a large reasoning token budget is quite important for getting good performance in such reasoning tasks.
>
> Our comparisons to GPT-5 Thinking are to contextualize the difficulty of our tasks, which we state in our paper. The token budget we give to GPT-5 is 10k tokens (stated in App. G), which we believe is reasonable. The token budget was never exceeded.
>
> > Also, being able to update the latent hypothesis at test time could be a huge advantage.
>
> Test-time latent hypothesis updates are what allow our model to achieve compositional generalization. That is a significant strength of our architecture, and we believe our results showing Abduction Transformer’s improved performance relative to models without test-time latent search only support the importance of the approach. We also note that test-time latent hypothesis updates on their own are not enough to achieve compositional generalization, which we show in our ablations against LPN (Macfarlane and Bonnet, 2025). In Table 1 and 2, we compare our model to LPN , which is an architecture that is also able to update its latent hypothesis at test-time. Our model, Abduction Transformer, surpasses its performance in 1-D ARC (25.1% ours vs 1.9% LPN) and SRAVEN (46.1% ours vs 37.1% LPN).
>
> > In table 2, why does the gradient-based refinement of H harms the solve rate when trained on 90% of the possible compositions?
>
> When trained on 90% of possible compositions on SRAVEN, Abduction Transformer achieves 96.4% +/- 0.4 with gradient search and 96.7% without. This is not a statistically significant difference, and can be attributed to noise.
>
> > In table 1 and 2, how does the performance of GPT scale with the reasoning token budget?
>
> The maximum token budget of 10k was never exceeded, so increasing this would have had no effect on performance.
>
> > How is the computational overhead compared to baseline methods such as Single Vector LPN?
>
> The primary additional computational overhead for Abduction Transformer comes from the fact that we perform test-time gradient search over sets of vectors as opposed to a single vector in the case of LPNs. Over 100 steps of test-time gradient search for 1-D ARC problems with batch size 512 on a single H100, Abduction Transformer executes 7.12 iterations/second and LPN executes 11.53 iterations/second.
>
> We thank the reviewer again for their useful feedback.

---

### Author Response · Authors · 2025-11-28
**Comments on Novelty**

**Novelty**

Reviewer 1Rbo expressed concerns about novelty. We believe this is an important point so we reiterate our response in a global comment.

While it is true that LPN introduces the idea of using a VAE for program synthesis with latent space search, our approach fundamentally differs in that LPN has a fixed-dimensional vector latent space, whereas our model’s latent space is a **set of weighted vectors whose size can grow with the required complexity**.  This fundamental difference leads to a number of novelties:
- LPN infers a distribution over vectors as a Gaussian, while we infer a distribution over mixture distributions (i.e. sets of weighted vectors) as a Dirichlet Process
- This enables our approach to represent compositional problems as sets of vectors whose size can vary according to the compositional complexity of the input, whereas LPN represents problems as a single, fixed size vector that cannot adapt to increased complexity
- The original LPN paper focuses on generic capabilities on program synthesis tasks, while we focus on explicitly measuring compositional generalization across abstract spatial, symbolic and linguistic tasks by testing on datasets containing verifiable compositions of those seen during training.
- Our approach performs significantly better on 1-D ARC for OOD compositions: 25.1% accuracy (ours) vs 1.9% accuracy (LPN), and 46.1% (ours) vs 37.1% (LPN) on SRAVEN.

---

### Author Response · Authors · 2025-12-03
**Summary of Reviewer Discussion Period**

Dear Chairs, thank you for your efforts in organizing this review process amidst unusual circumstances.

**We summarize our reviewer discussions below:**

A large part of the discussion period involved **reviewer VaZe and tPRa**, with reviewer tPRa raising issues with reviewer VaZe’s review, citing factual errors (please see tPRa’s comment ‘Use of Arbitrary Benchmarks Criticisms contains Factual Errors’). We thoroughly rebut reviewer VaZe’s concerns in our response to their initial review. Please see our responses to reviewer VaZe and subsequent discussion for important details.

**Reviewer 3Yws** recognized the strengths of our approach citing its strong OOD generalization on diverse tasks with a small number of parameters. They also recognized the improvement that comes from using nonparametric representations, as opposed to the fixed dimensionality representations used in LPN (Macfarlane and Bonnet, 2025). They raised the concern that we do not control for the effect of denoising attention on its own, but we clarified that we do in fact ablate denoising attention in our ‘Abduction Transformer (No gradient search)’ baseline in both §2 and §3. Their technical questions about GPT 5.1 token limit were addressed.

**Reviewer 1Rbo** raised concerns about the novelty of our approach compared to Latent Program Network (Macfarlane and Bonnet, 2025) and largely base their recommendation on this aspect. We thoroughly rebut their concerns in our response to their review and in our global comment ‘Comments on Novelty’. **Our results show 25.1% accuracy (ours) vs 1.9% accuracy (LPN) on 1-D ARC and 46.1% (ours) vs 37.1% (LPN) on SRAVEN.**

**Reviewer tPRa** asked for additional experiments on non-compositional datasets to verify that the advantage of our architecture over LPN truly stems from better compositional generalization and not other factors such as unrelated inductive biases or model capacity. We thought this was a great suggestion and presented a new set of results in exactly this setting, demonstrating that when LPN and Abduction Transformer (ours) are evaluated on problems NOT requiring compositional generalization, both LPN and Abduction Transformer perform equally well. (Please see response to tPRa for details.)

**Reviewer xg9b**’s comments and questions were thoroughly addressed in our response. Their concerns relating to objective, factual information, e.g., our KL approximation, denoising attention and our definition of composition, were clarified and substantiated by pointing to relevant sections already present in our paper and relevant literature that we cite. Their concerns about aspects relating to presentation (notation, phrasing) were helpful and addressed. 5/8 of the weaknesses raised by the reviewer fall into these two categories. Questions which were not of purely factual nature were also exhaustively addressed in our response. In their review, reviewer xg9b stated that “I am happy to update the score of the paper if my questions and concerns can be addressed.” We believe 5/8 of their concerns relating to objective aspects have been completely addressed. While we cannot be sure that the arguments we raised regarding the more subjective aspects of our experimental design would have definitively convinced the reviewer to raise their score, we have covered every concern (8/8) they have raised.

---

### Meta-Review · Area_Chair_vkbg · 2026-01-07

**Summary:**

The paper proposes the Abduction Transformer (AT), a novel architecture designed to improve compositional generalization in few-shot meta-learning tasks. The core innovation lies in modeling the latent hypothesis space as a nonparametric Dirichlet Process (DP) mixture over sets of vectors, rather than a single fixed-dimensional vector as seen in prior work like Latent Program Networks (LPN). The method employs variational inference with information-theoretic regularization and a test-time gradient-based search to refine latent hypotheses based on few-shot examples. The approach is evaluated on three tasks designed to test compositional generalization: 1-D ARC-like program induction, Symbolic Raven’s Progressive Matrices (SRAVEN), and linguistic systematicity.

Reason for Acceptance

The paper addresses a fundamental challenge in deep learning, systematic compositional generalization with a novel, principled approach. The authors effectively demonstrated that their nonparametric latent space significantly outperforms existing fixed-vector approaches on hard compositional tasks. They also successfully addressed legitimate reviewer concerns by providing new experiments that isolated the source of these gains. The negative reviews were largely based on either a debatable assessment of novelty (refuted by the performance gap) or a demand for benchmarks orthogonal to the paper's goals.

**Reviewer Concerns:**

Novelty Concerns (Reviewer 1Rbo): One reviewer argued the method lacks novelty compared to LPN. However, the move from parametric (Gaussian/Vector) to nonparametric (DP/Set-of-Vectors) is a distinct mathematical contribution. The order-of-magnitude improvement on the target metric (1.9% -> 25.1%) empirically validates that this is not merely an incremental tweak.

Benchmark Criticism (Reviewer VaZe): One reviewer strongly criticized the use of "niche" benchmarks (1-D ARC) and demanded ImageNet experiments. The meta-reviewer agrees that synthetic/symbolic tasks are appropriate and necessary for rigorously verifying compositional generalization, which is difficult to isolate in benchmarks like ImageNet.

Rigorous Evaluation of Compositionality: During the rebuttal, the authors provided a critical control experiment requested by Reviewer tPRa. They demonstrated that on non-compositional (in-distribution) tasks, AT performs comparably to LPN. This effectively isolates the performance gains to the model's ability to handle compositional generalization, verifying the central claim.

**Reviewer Scores:**

Reviewer 3Yws (Score: 8; likely no change): Supported the paper throughout based on the clear mechanism and strong results.

Reviewer tPRa (Score: 6 -> 8 likely): This reviewer requested the critical ablation on non-compositional data. Since the authors provided this and it supported the paper's claims, and given the reviewer's defense of the paper against unfair critiques, their score would likely have increased.

Reviewer xg9b (Score: 4 -> 5/6 likely): Raised valid technical questions (notation, convergence). The authors provided comprehensive clarifications that addressed the majority of these concerns.

Reviewer 1Rbo (Score: 2; likely no major change): Focused on novelty. While they might remain skeptical of the architectural leap, the empirical evidence provided in the rebuttal makes a "Reject" difficult to justify.

Reviewer VaZe (Score: 2; likely no change): Fundamentally disagreed with the problem setting (abstract reasoning vs. computer vision). This score is unlikely to change but is weighed less heavily as the critique targets the field rather than the specific contribution.

---

### Decision · Program_Chairs · 2026-01-26

Accept (Poster)